# Shared rhythmic subcortical GABAergic input to the entorhinal cortex and presubiculum

**Tim James Viney\*, Minas Salib, Abhilasha Joshi, Gunes Unal†, Naomi Berry‡, Peter Somogyi\***

Department of Pharmacology, University of Oxford, Oxford, United Kingdom

**Abstract** Rhythmic theta frequency (~5–12 Hz) oscillations coordinate neuronal synchrony and higher frequency oscillations across the cortex. Spatial navigation and context-dependent episodic memories are represented in several interconnected regions including the hippocampal and entorhinal cortices, but the cellular mechanisms for their dynamic coupling remain to be defined. Using monosynaptically-restricted retrograde viral tracing in mice, we identified a subcortical GABAergic input from the medial septum that terminated in the entorhinal cortex, with collaterals innervating the dorsal presubiculum. Extracellularly recording and labeling GABAergic entorhinal-projecting neurons in awake behaving mice show that these subcortical neurons, named orchid cells, fire in long rhythmic bursts during immobility and locomotion. Orchid cells discharge near the peak of hippocampal and entorhinal theta oscillations, couple to entorhinal gamma oscillations, and target subpopulations of extra-hippocampal GABAergic interneurons. Thus, orchid cells are a specialized source of rhythmic subcortical GABAergic modulation of 'upstream' and 'downstream' cortico-cortical circuits involved in mnemonic functions.

DOI: https://doi.org/10.7554/eLife.34395.001

**\*For correspondence:**
tim.viney@pharm.ox.ac.uk (TJV);
peter.somogyi@pharm.ox.ac.uk (PS)

**Present address:** †Boğaziçi Üniversitesi, Psikoloji Bölümü, Istanbul, Turkey; ‡Medical Research Council Brain Network Dynamics Unit, University of Oxford, Oxford, United Kingdom

**Competing interests:** The authors declare that no competing interests exist.

## Introduction

When an animal explores an environment, hippocampal 'place cells' represent different locations forming a spatial map and discharging rhythmic bursts of action potentials in temporal sequences (*O'Keefe and Recce, 1993*). Other spatially-modulated cells contribute to navigation, including 'head direction cells' in the presubiculum (PrS) (*Brandon et al., 2013*; *Taube et al., 1990*), and 'grid cells' and 'border cells' in the entorhinal cortex (EC) (*Hafting et al., 2005*; *Solstad et al., 2008*). Coordination of neuronal assemblies, such as place cell sequences, is controlled by local GABAergic neurons setting temporal windows of differential excitability and synchronizing neuronal activity over various time scales reflected by underlying network oscillations (*Cobb et al., 1995*; *Ylinen et al., 1995*). Such temporal coordination includes theta oscillations (~5–12 Hz) most prominent during movement and REM sleep (*Kramis et al., 1975*), gamma oscillations (~30–120 Hz), which are phase-amplitude coupled to theta (*Colgin et al., 2009*; *Lasztóczi and Klausberger, 2016*; *Schomburg et al., 2014*; *Soltesz and Deschenes, 1993*) and hippocampal sharp-wave associated ripple oscillations (SWRs, 130–230 Hz) mainly occurring during slow wave sleep, awake immobility and consummatory behavior (*Buzsáki, 1986*). Different GABAergic cell types target distinct subcellular domains of principal cells, each cell type preferentially firing during specific phases of network oscillations, which together contribute to a temporal redistribution of inhibition from the axon initial segment, to the soma, and all the way to the distal dendrites of principal cells (*Somogyi et al., 2014*; *Varga et al., 2012*). Given the diversity of rhythmic firing patterns these cortical GABAergic cell types, which neurons and circuits support their temporal coordination?

In the basal forebrain, the medial septum and the vertical and horizontal diagonal band nuclei (MSDB) contain cholinergic, glutamatergic and GABAergic neurons that project to different areas of the temporal cortex (*Kondo and Zaborszky, 2016*; *Manns et al., 2001*; *Unal et al., 2015*). The GABAergic projections preferentially target cortical GABAergic neurons in the hippocampus and retrosplenial cortex (*Freund and Antal, 1988*; *Freund and Gulyás, 1991*; *Unal et al., 2015*). One group of rhythmically-firing GABAergic MSDB neurons in the mouse named 'Teevra neurons' show selective innervation of the CA3 region of the hippocampus where they preferentially target axo-axonic cells and cholecystokinin-expressing (CCK) interneurons (*Joshi et al., 2017*). The functional roles of MSDB neurons have been investigated in vivo at the population level, with network effects mostly being studied in the dorsal CA1 (CA1d) region of the hippocampus, although most MSDB afferents to the hippocampus target CA3 and the dentate gyrus (*Freund and Antal, 1988*). Optogenetic activation of cholinergic MSDB neurons has been shown to promote theta oscillations in CA1d and dentate gyrus (*Mamad et al., 2015*; *Vandecasteele et al., 2014*). Glutamatergic MSDB neurons contribute to speed signals in both CA1d and medial EC (*Fuhrmann et al., 2015*; *Justus et al., 2017*), and subpopulations of GABAergic MSDB terminals in CA1d become activated during locomotion and salient stimuli (*Kaifosh et al., 2013*). *In vitro*, MSDB neurons have been shown to modulate the firing of hippocampal pyramidal cells and interneurons (*Huh et al., 2010*; *Leão et al., 2012*; *Tóth et al., 1997*). Recently, target neurons in the EC have been physiologically characterized based on their inputs from the MSDB, including those receiving parvalbumin (PV)-expressing input (*Fuchs et al., 2016*; *Gonzalez-Sulser et al., 2014*) and cholinergic input (*Desikan et al., 2018*). Other innervated cortical regions have received little attention (but see [*Unal et al., 2015*]). Does the area-selectivity of individual cholinergic MSDB neurons (*Wu et al., 2014*) and the combined area and synaptic target-neuron selectivity of single recorded and labeled GABAergic Teevra cells (*Joshi et al., 2017*) apply to other areas of the cortex innervated by the MSDB?

Neurons in the MSDB exhibit oscillatory firing at theta frequency, with rhythmically-firing neurons showing preferential coupling to different phases of theta cycles (*Dragoi et al., 1999*; *Joshi et al., 2017*; *King et al., 1998*; *Petsche et al., 1962*). Lesions or pharmacological inactivation of the MSDB result in impairments in spatial learning, a disruption of theta rhythmicity and grid cell firing dynamics in the EC, and a marked reduction in hippocampal theta power (*Brandon et al., 2013*; *Brito and Thomas, 1981*; *Hinman et al., 2016*; *Jeffery et al., 1995*; *Koenig et al., 2011*; *McNaughton et al., 2006*; *Yoder and Pang, 2005*). Subpopulations of GABAergic MSDB neurons, which include PV + Teevra neurons (*Joshi et al., 2017*) and other PV+ neurons (*Borhegyi et al., 2004*; *Simon et al., 2006*; *Varga et al., 2008*), likely represent some of the strong rhythmic bursting neurons that have been recorded in freely moving rats (*King et al., 1998*). It is currently unknown how different types of rhythmic bursting neurons contribute to cortical circuits outside the CA3 area (*Joshi et al., 2017*). This is due to the wide range of preferred theta phase coupling by individual cells, the general lack of information on the topography of MSDB projections covering both the hippocampal formation and extra-hippocampal cortical regions, and until recently, a lack of information on single long-range projection axons. We have investigated the behavior-dependent firing patterns, axonal projections and cortical targets of GABAergic MSDB neurons that projected to the EC in awake mice.

## Results

### Medial septal neurons projecting to the entorhinal cortex

We investigated MSDB projections to the EC by injecting a monosynaptically-restricted retrograde viral tracer, PRV-hSyn-Cre (*Oyibo et al., 2014*), into the *caudo-dorsal* EC (*Figure 1—figure supplement 1a,b*) and a Cre-dependent adeno-associated virus (AAV) encoding EYFP into the MSDB. The PRV-hSyn-Cre virus, a mutant pseudorabies virus of the alpha-herpesvirus subfamily, is non-cytotoxic and highly tropic, resulting in Cre expression only in neurons that directly project to the injection site. After ≥2 weeks incubation, strong Cre-dependent EYFP expression in neurons was mainly in the rostral part of the dorsal medial septum (MS; *Figure 1a*, *Figure 1—figure supplement 1c*) representing 60.5% of retrogradely-labeled neurons (n = 129 cells from six mice; mean ± s.d. 21.5 ± 11.7 total EYFP+ neurons/mouse). The remainder were distributed in the vertical DB (24.8%), horizontal DB (12.4%) and lateral septum (2.3%).

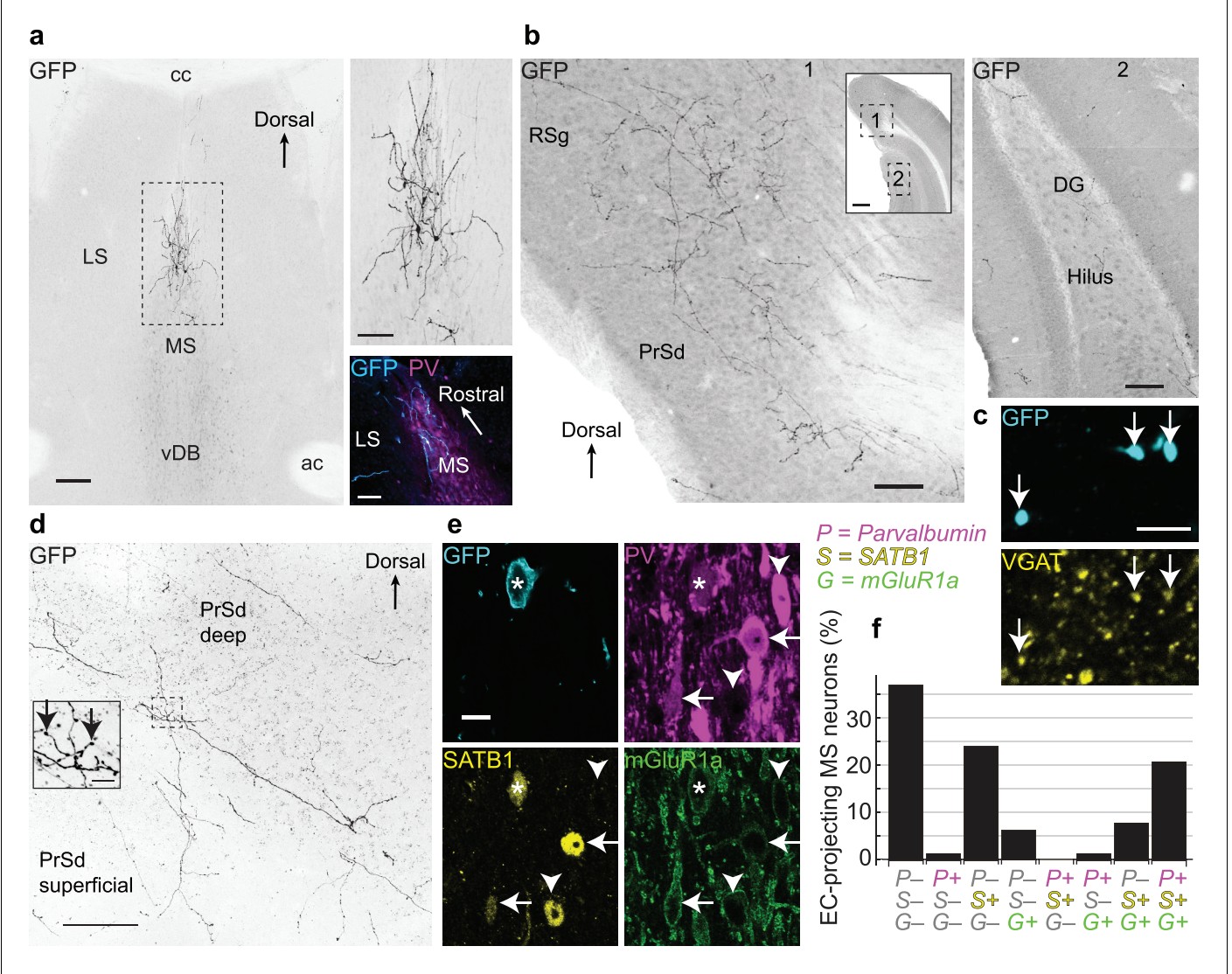

**Figure 1.** Medial septal GABAergic neurons terminating in the entorhinal cortex also innervate the dorsal presubiculum and retrosplenial cortex. (a) Coronal section showing EC-projecting GFP-immunoreactive neurons restricted to the dorsal MS, following injection of PRV-hSyn-Cre into the caudo-dorsal EC and AAV$^{DIO-EYFP}$ into the MSDB (animal MS60). Top right, enlarged view of the boxed region. Bottom right, horizontal section (animal MS77) showing GFP-immunoreactive neurons (cyan) restricted to the rostral part of the dorsal MS, delineated by PV immunoreactivity (magenta). (b) Virally-labeled axon collaterals of EC-projecting MSDB neurons densely innervating extra-hippocampal regions (left, 1) and sparsely innervating the DG and CA1 (right, 2) from a single coronal section (animal MS60). Inset, locations of *Figures 1* and *2* (boxes). (c) A subset of axon terminals from EC-projecting MSDB neurons in the RSg (GFP, cyan) are immunoreactive for VGAT (yellow, arrows) (animal MS66). (d) Coronal section of the PrSd (animal MS60) with axon collaterals and terminals from EC-projecting MSDB neurons. Inset, enlarged view of boxed region (arrows, axon terminals). (e) An EC-projecting medial septal neuron soma (asterisk, from animal MS66) immunoreactive for GFP (cyan, plasma membrane) was weakly immunoreactive for PV (magenta), SATB1 (yellow, nucleus) and mGluR1a (green, membrane). Note GFP-negative neurons with similar (arrows) or different (arrowheads) molecular profiles. (f) Quantification of PV (P), SATB1 (S) and mGluR1a (G) immunoreactivity for virally-labeled EC-projecting neurons located in the dorsal MS (data from six animals). Scale bars (µm): (a) 200 (left image), 100 (right images); (b) 100 (inset 500); (c) 5; (d) 100, inset 10; (e) 10. Image type: (a–b) Widefield epifluorescence, reverse contrast, 70 µm thick sections. (c) confocal image, single optical section, 0.31 µm thick; (d) confocal image, reverse contrast, maximum intensity z-projection, 35 sections, 30.96 µm thick; (e) confocal image, single optical section, 0.38 µm thick. LS, lateral septum; cc, corpus callosum; ac, anterior commissure.

DOI: https://doi.org/10.7554/eLife.34395.002

The following figure supplement is available for figure 1:

**Figure supplement 1.** Medial septal neurons terminating in the entorhinal cortex also innervate the dorsal presubiculum.
DOI: https://doi.org/10.7554/eLife.34395.003

Virally labeled medial septal axons with extensive terminals were observed in all layers of the EC (n = 4 mice, from 18 coronal or horizontal sections), with collaterals traveling both radially and horizontally (*Figure 1—figure supplement 1a,b*). These EC-projecting axons also gave dense collaterals in other extra-hippocampal regions where they formed terminals (n = 6 mice; *Figure 1b–d*, *Figure 1—figure supplement 1d*). We quantified the distribution of axonal collaterals in hippocampal and extra-hippocampal cortical regions (n = 3 mice; three sections per animal; total 274 axons). The proportion of axonal branches in the dorsal presubiculum (PrSd) and granular retrosplenial cortex (RSg) (median: 37%; interquartile range (IQR): 34.3–47.2%) was substantially greater than in CA1, the dentate gyrus and the dorsal subiculum (SUBd) (median: 5%; IQR: 2.4–9%), where only rare axonal collaterals were observed (p=1.6 $\times$ 10$^{-7}$, Kruskal-Wallis test). Within the PrSd and RSg, 71.0 ± 25.9% of EYFP+ axonal terminals (mean ±s.d, n = 1046/1416 counted terminals within 12 sampled areas from three mice) were immunoreactive for vesicular GABA transporter (VGAT, *Figure 1c*). Together these results demonstrate that a large subpopulation of GABAergic EC-projecting MSDB neurons innervate other extra-hippocampal areas, primarily the PrSd and RSg.

Neuronal subpopulations in the MSDB can be defined by the expression of different molecules (*Wei et al., 2012*), and combinational expression profiles help define distinct cell types (*Viney et al., 2013*). We observed that metabotropic glutamate receptor 1a (mGluR1a), along with the transcription factor SATB1 (*Huang et al., 2011*), show differential immunoreactivity with parvalbumin (PV). As in the cortex, PV neurons in the MSDB represent a subpopulation of GABAergic neurons, but PV is expressed by many different kinds of neurons (*Simon et al., 2006*; *Varga et al., 2008*; *Viney et al., 2013*). GABAergic Teevra neurons are immunopositive (+) for PV *and* SATB1 but lack detectable immunoreactivity (–) for mGluR1a (*Joshi et al., 2017*). Within the MS, 21.0% of EC-projecting (EYFP +) neurons were triple immunopositive for PV, SATB1, and mGluR1a, while 37.1% showed no detectable signal for these three molecules (n = 62 tested neurons, six mice; *Figure 1e,f*). Neurons found nearby as PV+/SATB1+/mGluR1a– did not project to the EC (*Figure 1f*), consistent with the profile of CA3-projecting Teevra cells (*Joshi et al., 2017*). Within the DB, 44.8% of neurons were only SATB1+ with the remainder of cells being triple immunonegative (n = 29 tested neurons, six mice). A subset of EYFP+ neurons were tested for choline acetyltransferase (ChAT) (*Kondo and Zaborszky, 2016*; *Unal et al., 2015*), PV and mGluR1a. Only 2/13 tested were ChAT+/PV–/mGluR1a–, 1/13 was only mGluR1a+, and 10/13 lacked detectable immunoreactivity (ChAT–/PV–/mGluR1a–). The PV– and ChAT– MSDB neurons probably comprise both GABAergic and glutamatergic EC-projecting neurons (*Fuchs et al., 2016*; *Gonzalez-Sulser et al., 2014*; *Justus et al., 2017*).

## Firing patterns of septo-entorhinal neurons

To determine the temporal patterns of signals sent from the basal forebrain to the EC, we lowered glass electrodes into the MSDB in drug-free awake head-restrained mice to record neurons extracellularly, then visualized their axon collaterals and terminals in the cortex by juxtacellular labeling (*Figure 2*). Eight labeled neurons had projection axons heading towards the EC via the dorsal fornix, with 5/8 forming thin collaterals and terminals within deep layers of the caudo-dorsal EC (*Figure 3*, *Figure 3—figure supplement 1*, *Tables 1* and *2*). The main axons of the other three neurons faded before collateralization in the EC due to insufficient labeling (*Table 1*). Nevertheless, all eight neurons showed strong rhythmic burst firing occurring during both locomotion and immobility (*Figures 2a,b,d,e* and *3f,g*), with longer bursts displaying accommodation (*Figures 2a,b* and *3f, g*). Approximately 10% of bursts were >200 ms, and were often associated with high gamma power in hippocampal LFPs during immobility (*Figures 2a,b* and *3g*) (*Carr et al., 2012*).

We simultaneously recorded the local field potential (LFP) in strata pyramidale/oriens of the dorsal CA1 (CA1d) region of the hippocampus, a reliable and stable reference location for comparing different neurons. All neurons were significantly coupled to theta oscillations, with 6/8 preferentially firing around the peak, and 2/8 on the descending phase (p<0.0001 for all cells, Rayleigh test for uniformity; *Figures 2a,b,f* and *3f*, *Table 1*). Cortical principal and GABAergic neurons couple to gamma oscillations nested within theta cycles, with ~55–80 Hz 'mid-gamma' oscillations arriving in CA1 directly from the EC around the peak of pyramidale/oriens theta cycles (*Colgin et al., 2009*; *Harris et al., 2003*; *Lasztóczi and Klausberger, 2016*; *Schomburg et al., 2014*; *Soltesz and Deschenes, 1993*; *Somogyi et al., 2014*). As expected, mid-gamma oscillations were coupled to theta cycles preferentially at the theta peak (detected from CA1d LFPs of n = 7/8 animals with recorded septo-entorhinal neurons, *Figure 2g*). Mid-gamma oscillations were also detected during

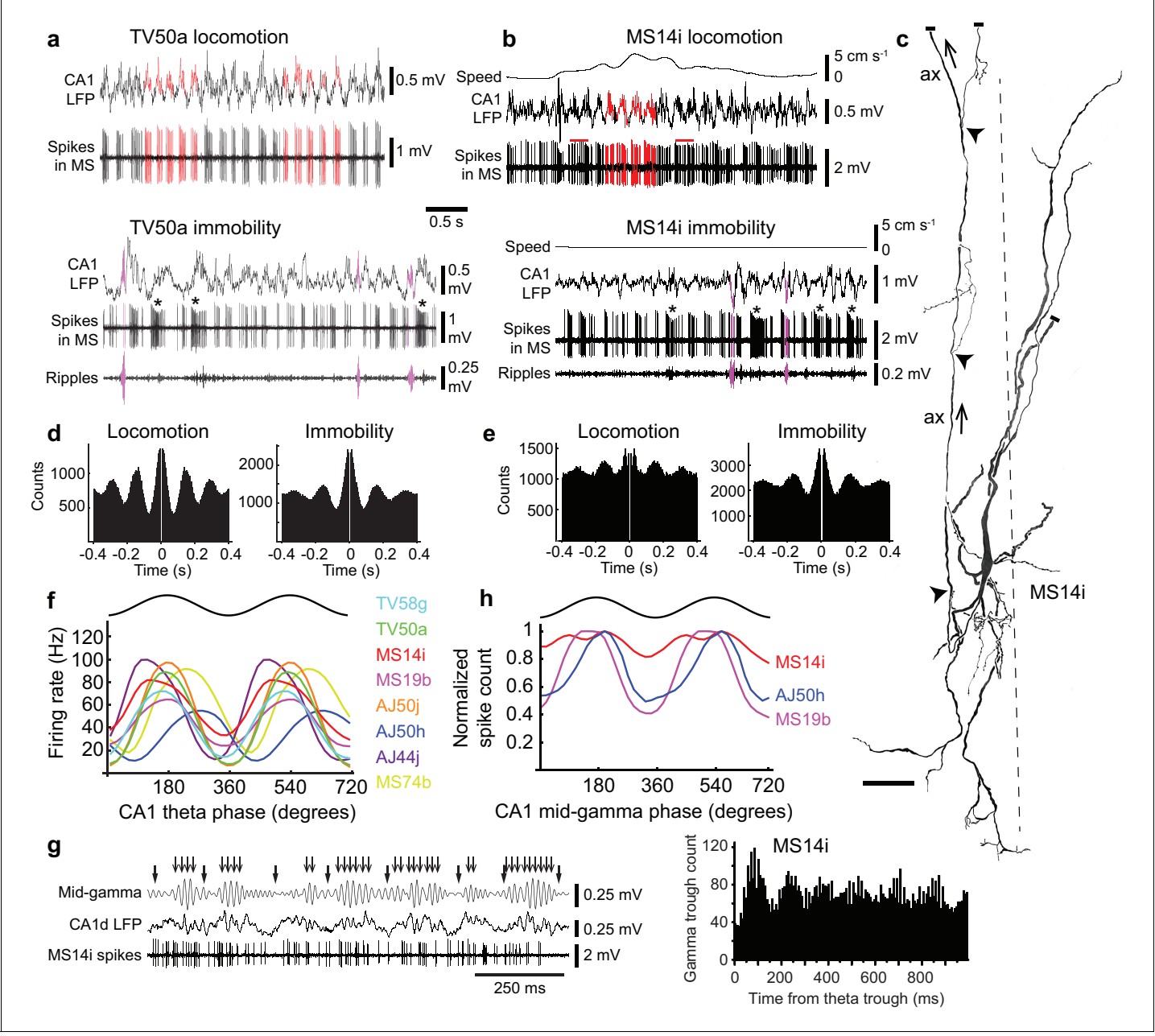

**Figure 2.** Firing patterns of medial septal neurons projecting to the EC. (a) Top, rhythmic burst firing of EC-projecting medial septal neuron TV50a during locomotion. CA1 LFP, local field potential in SP. Red, examples of bursts of spikes associated with peaks of CA1 theta oscillations. Bottom, rhythmic burst firing during immobility with some spikes during some SWRs (magenta; 130–230 Hz filtered CA1 LFP). Examples of long accommodating bursts are shown below the asterisks. (b) Same as in (a) for another EC-projecting neuron, MS14i, which fires less rhythmic bursts at high rate during locomotion (top) and rhythmic bursts during immobility (bottom). Red bars, examples of bursts spanning two theta cycles. (c) Reconstruction of the soma, dendrites, and axon (ax; arrows: towards dorsal fornix) of MS14i in the tip of dorsal MS. Dashed line, midline. Local axon collaterals, which give rise to terminals, are marked by arrowheads. Cut ends not included in the reconstruction are marked by small bars. Scale bar, 50 μm. (d,e) Autocorrelograms of spikes during locomotion and immobility for TV50a (d) and MS14i (e). (f) Firing rate versus CA1 theta phase plots show preferential firing near the peak of theta during movement (locomotion and small movements) for eight labeled neurons (color-coded) projecting to extra-hippocampal regions. Black, sinusoid representation of two idealized cycles. Data are duplicated to represent rhythmicity. (g) Left, part of the time series from (a) during locomotion displaying several theta cycles in the unfiltered LFP (thick arrows mark theta troughs), and 55–80 Hz filtered LFP mid-gamma cycles (thin arrows mark high amplitude gamma troughs). Note that gamma troughs occur mostly at the peak of theta cycles. Right, event correlation of detected mid-gamma troughs relative to theta troughs from the MS14i recorded CA1d LFP. Note peak around 100 ms corresponding to the peak of theta cycles. (h) Normalized spike count versus CA1 mid-gamma phase for all significantly coupled EC-projecting neurons (see also *Table 3*).

*Figure 2 continued on next page*

*Figure 2 continued*

DOI: https://doi.org/10.7554/eLife.34395.004

The following figure supplement is available for figure 2:

**Figure supplement 1.** Rhythmicity of identified orchid cells.

DOI: https://doi.org/10.7554/eLife.34395.005

'large amplitude irregular activity' (LIA) of the LFP (between periods of high power theta oscillations, also referred to as 'non-theta'; *Figures 2a,b* and *3g*). A subset of neurons preferentially coupled to the peak of mid-gamma oscillations (p<0.001 for n = 3/7 tested neurons, Rayleigh test; *Figure 2h*, *Table 3*), suggesting that theta-coupled septo-entorhinal neurons participate in entorhinal cortical gamma activity (*Chrobak et al., 2000*).

## GABAergic orchid cells project to multiple extra-hippocampal regions

As predicted by the viral tracing, single recorded and labeled EC-projecting neurons innervated multiple extra-hippocampal regions, with the majority of their terminals outside the EC located in the PrSd (*Figures 3* and *4*, *Figure 3—figure supplement 2*, *Figure 3—figure supplement 3*, *Figure 3—figure supplement 4*, *Table 1*, *Figure 3—video 1*). Extracellularly recorded EC-projecting neurons were immunopositive for PV (n = 8/8 tested cells), SATB1 (n = 6/6), and mGluR1a (n = 6/6), and their terminals were immunoreactive for VGAT (n = 2/2 tested cells; *Figure 3b,d,e*, *Table 4*). Based on the shared cortical target regions (*Table 1*), similarity in rhythmic burst firing patterns (*Figure 2a,b*, *Figure 3f,g*; see below), strong theta coupling (*Figure 2f*), overlapping molecular profiles (*Table 4*), and the resemblance of the axon to *Phalaenopsis* orchids (*Figure 3—figure supplement 5*), we name these GABAergic PV+ neurons 'orchid cells'.

Within the dorsal tip of the MS, orchid cell dendritic trees had prominent apical and basal dendrites (*Figures 2c* and *3c*). Dendrites were often twisted, exhibited filopodia, with the basal dendrites ending in specialized filopodia-like distal tips (*Figures 3c* and *2c*). Orchid cells had 6.0 ± 1.8 primary dendrites (mean ±s.d., n = 4 cells) extending from all axes, which were similar to the distributions of CA3-projecting Teevra cell dendrites (*Joshi et al., 2017*) ($\chi$2 = 0.02, p=0.8817, n = 4 orchid cells versus n = 4 Teevra cells). The hooked (e.g. MS14i, AJ50h, AJ50j) or straight (e.g. TV58g) projection axons could originate from the soma, the basal (MS14i, *Figure 2c*) or apical (TV58g, *Figure 3c*) dendrite, with local collaterals forming terminals within the MS. Basket-like groups of axon terminals were observed around other somata. We conclude that PV+/SATB1+/mGluR1a + orchid cells provide rhythmic bursts of GABAergic input to specific extra-hippocampal regions of the temporal cortex.

## Preferential synaptic targets of orchid cells

To gain insight into the role of rhythmic GABAergic input from orchid cells to cortical circuits, we followed orchid-like collaterals to synaptic targets (*Figures 3a* and *4*). Terminals of strongly-labeled orchid cell TV58g were observed in the *fasciola cinereum* (FC; marked by *Amigo2* mRNA expression as in CA2) (*Figure 3a*, *Figure 3—figure supplement 4a,b*, *Table 5*; see also Materials and Methods - Additional details of orchid cell projections) (*Laeremans et al., 2013*), followed more temporally by terminals in RSg layer 6 (*Figure 3a*, *Figure 3—figure supplement 2b*); this region was also innervated by another cell (AJ50j, *Table 1*). Apart from the FC, no terminals of any orchid cell were observed in the hippocampus or dentate gyrus (n = 0/6 cells, *Table 1*). We observed that the PrSd could be defined by *Slc17a6*, *Satb2*, *Etv1* and *Nos1* expression patterns (*Figure 3—figure supplement 2*, *Figure 3—figure supplement 3*). This region was extensively innervated by TV58g via three main collaterals, a smaller branch in the angular bundle and a minor sub-branch extending from the PrSd into the SUBd (*Figures 3a* and *4a*). Terminals were exclusively in PrSd layers 5–6, defined by differential calbindin immunoreactivity (41.6% of cortical varicosities, *Table 1*, *Figure 3—figure supplement 3d*); other cells additionally innervated more superficial layers (n = 5 cells, *Table 1*). Interestingly, axon terminals were found at high density in PrSd 'hot spots', such as in the extreme medial zone (*Figure 4a*). The SUBd was also innervated by three other cells (*Table 1*). The axon of TV58g subsequently extensively innervated caudo-dorsal EC (45.6% of cortical varicosities, *Figure 3a*, *Figure 3—figure supplement 1*, *Figure 3—figure supplement 3*, *Table 1*) preferentially

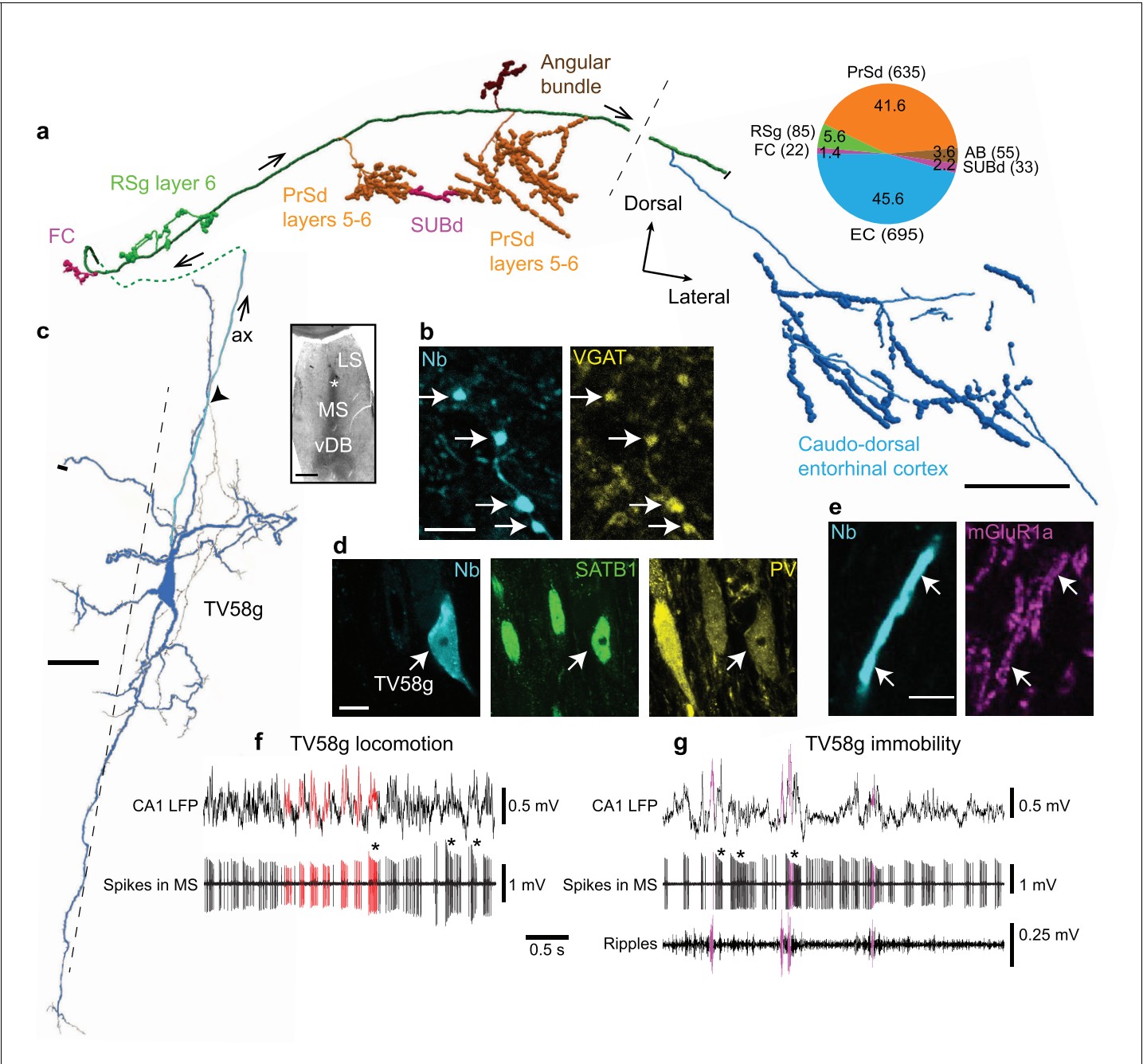

**Figure 3.** Firing patterns and cortical target regions of a GABAergic medial septal orchid cell. (**a**) Reconstruction of part of the axon of neuron TV58g axon (cortical regions color-coded). Black dashed line, rostral (left) and caudal (right) innervation zones. Pie chart, number (parentheses) and proportion (%) of axonal varicosities. Projection axon enters the cortex via the dorsal fornix (green dashed line). (**b**) Presubicular terminals of TV58g (cyan) were VGAT+ (yellow, arrows). (**c**) Micrograph (left, asterisk) and reconstruction (right) of the soma and dendrites (blue) and axon (cyan, ax) of TV58g in the tip of dorsal MS. Black dashed line, midline. A local axon collateral is marked by an arrowhead. A cut dendrite is marked by a small bar. (**d**) The soma of TV58g (cyan, neurobiotin, arrow) was SATB1+ (green, nuclei) and PV+ (yellow). Note similar profiles of adjacent neurons. (**e**) Dendrites of TV58g (cyan) were mGluR1a+ (magenta, arrows). (**f**) Rhythmic burst firing during locomotion. CA1 LFP, local field potential in SP. Red, examples of detected peaks of CA1 theta oscillations and associated MS neuron spikes. Note preferential coupling of bursts to theta peak. (**g**) Rhythmic burst firing during immobility with some spikes during some SWRs (magenta; ripples, 130–230 Hz filtered CA1 LFP). Examples of accommodating bursts are shown below asterisks in (**f**) and (**g**). Nb, neurobiotin. FC, fasciola cinereum, AB, angular bundle, LS, lateral septum, vDB, vertical diagonal band. Images (z-thickness in μm and z-projection type): (**d**) 2.3, maximum, (**b**) 2.61, maximum, (**e**) 0.37, single optical section. Scale bars (μm): a, 500, b,e, 5, c, 50, inset, 500, d, 10.
DOI: https://doi.org/10.7554/eLife.34395.008

The following video and figure supplements are available for figure 3:

*Figure 3 continued on next page*

*Figure 3 continued*

**Figure supplement 1.** Location of the axons of orchid cells in the caudo-dorsal entorhinal cortex.
DOI: https://doi.org/10.7554/eLife.34395.009
**Figure supplement 2.** Location of labeled orchid cell axon in the dorsal presubiculum (I).
DOI: https://doi.org/10.7554/eLife.34395.010
**Figure supplement 3.** Location of labeled orchid cell axon in the dorsal presubiculum (II).
DOI: https://doi.org/10.7554/eLife.34395.011
**Figure supplement 4.** Delineation of synaptic target regions of orchid cells.
DOI: https://doi.org/10.7554/eLife.34395.012
**Figure supplement 5.** Resemblance of a medial septal axon to a moth orchid.
DOI: https://doi.org/10.7554/eLife.34395.013
**Figure 3—video 1.** Reconstruction of orchid cell TV58g.
DOI: https://doi.org/10.7554/eLife.34395.014

in layer 6 (expression of the *Tmem178* gene; *Figure 3—figure supplement 4c*, *Table 5*); we observed terminals from three other orchid cells at a similar location (*Figure 3—figure supplement 1*, *Table 1*). No terminals were observed in the *Wfs1*-expressing PaS region (n = 0/6 cells; *Table 1*; *Figure 3—figure supplement 4d*).

We determined the molecular profiles of 13 targeted cortical neurons from the most strongly-labeled orchid cell (n = 1 neuron; *Table 6*) and used electron microscopy to test the reliability of predicting synaptic connections from light microscopy. In the PrSd, 5/5 tested targets were PV+ (*Figure 4a,b*) and 3/7 tested were nNOS + interneurons, including 1 PV+ cell (*Table 6*). The PrSd expresses a high level of *Nos1* mRNA (*Figure 3—figure supplement 2d*), and we found ~70% overlap of PV and nNOS immunoreactivity (data not shown). In the electron microscopic study (from the same orchid cell), all 11 tested boutons formed synapses with interneurons; 6 (55%) of them dendrites (*Figure 4ci,ii*), and five somata (*Figure 4ciii*), including a PV+/SATB1+/mGluR1a + interneuron (*Figure 4b*) receiving four synapses (*Figure 4ci*). These data are consistent with postsynaptic interneuron targets of GABAergic MS terminals in rodent CA3 and RSg (*Freund and Antal, 1988*; *Freund and Gulyás, 1991*; *Joshi et al., 2017*; *Unal et al., 2015*; *Viney et al., 2013*). Overall, somata and proximal dendrites of GABAergic interneurons received extensive innervation by orchid cell terminals (*Figure 4b–f*). Targets in three regions (PrSd, RSg and FC) were SATB1+ (n = 13/13; *Figure 4b,e*), but CCK– (n = 0/10) (*Table 6*). The axon showed target selectivity, as nearby interneurons with different molecular profiles were not targeted (*Figure 4b*). The identity of SATB1+ neurons in these regions remain to be determined by visualizing the processes of the cells, and are likely to include both PV+ and somatostatin+ interneurons (*Nassar et al., 2015*), given that SATB1 is highly expressed within these subpopulations in the cortex (*Close et al., 2012*; *Denaxa et al., 2012*). Full characterization of GABAergic interneurons in the PrS, RSg, and EC awaits further studies possibly using transcriptomic data. We observed for four orchid cells that their axon terminals were apposed to some cortical neurons with high levels of endogenous biotin (e.g. *Figure 4d*; n = 8/13 targets, *Table 6*), suggesting that these are interneurons with high metabolic activity, such as 'fast spiking' neurons. These data, along with the preferential targets of PV+ MSDB axons in EC (*Unal et al., 2015*), suggest that PV+ orchid cells influence extra-hippocampal regions, via rhythmic modulation of select subpopulations of local GABAergic interneurons innervating them (*Figure 4g*).

## Behavioral and network state-dependent firing of orchid cells

Dynamic changes in network oscillations reflect ongoing behavior, with marked changes in power often indicating a transition to a different behavioral state. We categorized four behavioral states in the head-restrained mouse: locomotion (LM); small movements (SM), which typically included postural shifts, limb and tail motion; spontaneous whisking and/or sniffing (WS) without other movements; and immobility (IM) (*Table 7*). During LM and SM, theta oscillations dominated the CA1d LFP (*Figures 2a,b*, *3f* and *5a*). During WS and IM, LIA emerged in the hippocampus and more regular 'delta' 2–4 Hz oscillations were present in the MS (*Figure 5—figure supplement 1a*) (*Fujisawa and Buzsáki, 2011*; *Nerad and McNaughton, 2006*; *Wolansky et al., 2006*), often at higher power when the mouse was breathing more deeply. Orchid cell rhythmicity was evident during all four behavioral states, with autocorrelograms exhibiting robust peaks and troughs (*Figure 2d,e*,

**Table 1.** Axon terminal distribution and firing patterns of identified orchid cells.

All projection axons traveled in the dorsal fornix. *Varicosities were sampled from different numbers of brain sections per labeled neuron. **Main axon observed dorsal of SUBd/PrSd, too weak to observe collaterals. ***Main axon faded away just rostral of caudo-dorsal EC. Abbreviations: c, fine axon collaterals observed; u, unknown/unavailable; n.o., not observed. Spike burst defined as >3 spikes with interspike intervals (ISIs) < 40 ms. Firing rates and burst incidence (Hz) are expressed as mean of 1 s bins ± s.d.; intraburst frequency (Hz) as mean ±s.d.; burst duration and interburst interval (ms) are expressed as median and interquartile range. Abbreviations: LM, locomotion; SM, small movements including limbs, tail, and shifts in posture; WS, high-frequency whisking and/or sniffing in the absence of other movements; IM, immobility. KS, Kolmogorov-Smirnov; SP, stratum pyramidale; SO, stratum oriens.

| Cell name | | TV58g | TV50a | MS14i | MS19b | AJ50j | AJ50h | AJ44j | MS74b |
|---|---|---|---|---|---|---|---|---|---|
| Target hemisphere | | Right | Right | Left | Left | Left | Right | Right | Right |
| Varicosities per region (n sampled)* | Medial septum | 61 | u | 146 | n.o. | n.o. | 51 | n.o. | u |
| | Dentate gyrus | n.o. | n.o. | n.o. | n.o. | n.o. | n.o. | n.o. | u |
| | CA3 | n.o. | n.o. | n.o. | n.o. | n.o. | n.o. | u | u |
| | CA2 | n.o. | n.o. | n.o. | n.o. | n.o. | n.o. | u | u |
| | CA1 | n.o. | n.o. | (c) | n.o. | n.o. | n.o. | u | u |
| | SUBd | 33 | n.o. | 3 | 104 | n.o. | c | u | u |
| | FC | 22 | (c) | n.o. | n.o. | n.o. | n.o. | u | u |
| | RSg | 85 (L6) | n.o. | n.o. | n.o. | 51 | n.o. | u | u |
| | Angular Bundle | 55 | 11 | n.o. | n.o. | n.o. | n.o. | u | u** |
| | PrSd L1-2 | n.o. | 3 | 8 | 15 | c | n.o. | u | u |
| | PrSd L3 | n.o. | 10 | 31 | 149 | n.o. | c | u | u |
| | PrSd L5-6 | 635 | 66 | 45 | 34 | 27 | 29 | u | u |
| | PaS | n.o. | n.o. | n.o. | n.o. | n.o. | n.o. | u | u |
| | EC | 695 | c | 32 | 63 | (c)*** | 6 | u | u |
| Labeling strength | | +++ | ++ | ++ | ++ | ++ | ++ | + | + |
| Behavioral states | Max speed (cm/s) | u | u | 6.3 | 18.8 | 17.2 | 16.5 | 12.0 | u |
| | Firing rate LM | 48.4 ± 6.0 | 51.4 ± 10.6 | 67.8 ± 12.9 | 43.4 ± 9.3 | 65.3 ± 11.7 | 33.6 ± 9.6 | 76.7 ± 11.0 | 55.0 ± 11.0 |
| | Firing rate SM | 41.6 ± 5.3 | 47.6 ± 8.8 | 51.0 ± 12.7 | 33.3 ± 8.6 | 42.5 ± 9.7 | 27.7 ± 3.8 | 43.8 ± 9.9 | u |
| | Firing rate WS | 47.1 ± 9.3 | u | 47.9 ± 9.0 | 33.2 ± 8.0 | 34.7 ± 6.1 | 24.3 ± 4.9 | 37.7 ± 2.5 | 34.2 ± 14.9 |
| | Firing rate IM | 44.2 ± 12.0 | 46.3 ± 10.5 | 48.2 ± 10.0 | 31.9 ± 5.2 | 46.6 ± 9.9 | 25.4 ± 7.6 | 38.1 ± 6.8 | 37.6 ± 16.0 |
| | Burst incidence LM | 5.0 ± 1.0 | 6.3 ± 0.9 | 3.8 ± 1.7 | 4.4 ± 0.9 | 6.7 ± 1.0 | 4.1 ± 1.7 | 6.2 ± 1.9 | 5.2 ± 1.3 |
| | Burst incidence SM | 4.4 ± 0.5 | 5.7 ± 1.0 | 4.7 ± 1.1 | 3.6 ± 1.1 | 5.5 ± 1.3 | 3.3 ± 0.8 | 5.6 ± 1.2 | u |
| | Burst incidence WS | 4.4 ± 0.9 | u | 4.6 ± 1.1 | 4.4 ± 1.3 | 4.6 ± 1.0 | 2.8 ± 0.8 | 6.0 ± 1.0 | 2.9 ± 1.4 |
| | Burst incidence IM | 4.2 ± 1.2 | 5.0 ± 0.9 | 4.9 ± 0.9 | 4.5 ± 1.3 | 5.4 ± 1.0 | 3.1 ± 1.3 | 5.3 ± 1.1 | 3.5 ± 0.9 |
| CA1d theta (5 – 12 Hz) | LFP measurement | SO/SP | SP | upper SP | SP | SP | SP | SP | SP |
| | Preferred theta phase | 159° | 176° | 152° | 167° | 177° | 264° | 130° | 233° |
| | Mean vector length (r) | 0.37 | 0.50 | 0.23 | 0.27 | 0.51 | 0.35 | 0.44 | 0.34 |
| | Rayleigh P-value | <0.0001 | <0.0001 | <0.0001 | <0.0001 | <0.0001 | <0.0001 | <0.0001 | <0.0001 |
| | Total n spikes | 1081 | 2679 | 6659 | 5143 | 1743 | 1413 | 860 | 578 |
| | Spikes per cycle (mean ± s.d.) | 6.8 ± 3.1 | 6.7 ± 3.0 | 8.7 ± 4.2 | 7.2 ± 3.4 | 7.2 ± 3.1 | 4.6 ± 1.9 | 7.7 ± 3.3 | 8.1 ± 3.2 |

DOI: https://doi.org/10.7554/eLife.34395.006

Figure 2—figure supplement 1). The peak in the autocorrelograms was at 143.8 ± 19.8 ms during locomotion (mean ±s.d., n = 8 cells) but occurred later during the other states (SM: 169.6 ± 38.8 ms, n = 7; WS: 167.5 ± 18.6 ms, n = 6; IM: 174.4 ± 26.4 ms, n = 8; *Figure 2—figure supplement 1*). Rhythmic firing was thus within the lower frequency range of 5–12 Hz theta oscillations (~80–200 ms) during all behavioral states, with cells exhibiting faster rhythmic firing patterns during locomotion when theta oscillations were at their highest frequency and power (*Figure 5a, Figure 2—figure*

**Table 2.** Details of juxtacellular labeling.

Number of juxtacellular labeling attempts on the final recording day for each animal (neurobiotin is metabolized within 24 hr). Letters refer to individually recorded neurons. E.g. in animal TV58, recorded neurons 'e' and 'g' were selected for juxtacellular labeling; in animal TV50 only neuron 'a' was recorded and labeled. A 'labeling attempt' refers to the application of current pulses to a recorded neuron. Modulation refers to successful entrainment of action potentials to the duration of positive current pulses (*Pinault, 1996*). Strength and duration of modulation are also estimated in order to predict the labeling strength (*Table 1*).

| Animal name | Labeling attempts from single neurons | Modulated neurons | Recovered neurons |
|---|---|---|---|
| TV58 | 2 (e, g) | 1 (g, very strong, >5 min; rostral of e) | 1 (g) |
| TV50 | 1 (a) | 1 (a, strong, 77 s) | 1 (a, damaged) |
| MS14 | 1 (i) | 1 (i, strong, 1 min) | One strong (i); one weak, dorsal of i, dendrites cross with i, potentially gap-junction coupled |
| MS19 | 1 (b) | 1 (b, strong, 30 s) | 1 (b) |
| AJ50 | 4 (g-j) | 3 (g, strong, 6 s, lost; h strong 30 s, dorsal of g; j, strong, 2 min, rostral of h) | Two strong (h, j); one very weak; dorsal of h, potentially gap-junction coupled |
| AJ44 | 1 (j) | 1 (j, strong, 12 s) | 1 (j) |
| MS74 | 1 (b) | 1 (b, strong, 2 min) | 1 (b, damaged) |
| MS17 | 1 (k) | 1 (k, strong, 30 s, lost cell after) | 1 (k) |

DOI: https://doi.org/10.7554/eLife.34395.007

supplement 1). Theta oscillations were observed intermittently during SM, WS and IM, which may account for orchid cell rhythmic firing at the lower theta frequency range. But when we tested for rhythmic burst firing during CA1d LIA (<5 Hz) 'non-theta' periods (*Figure 5—figure supplement 1a*), we found that 5/8 orchid cells increased firing close to the peak of the variable duration LIA cycles, with bursts beginning on the rising phase (neurons AJ44j, TV50a, TV58g, AJ50j, and MS19b; *Figure 5b*). These firing patterns are consistent with the preferential firing around the peak of each theta cycle (*Figure 2f*). These data suggest that orchid cells can couple to irregular low frequency (<5 Hz) cortical activity in addition to 5–12 Hz theta oscillations.

In addition to behavioral state-dependent differences in rhythmic firing, the mean firing rates of orchid cells during locomotion were different than rates during immobility (mean ±s.d. 55.2 ± 14.1 Hz LM versus 39.8 ± 8.1 Hz IM; n = 8, $t_7$ = 3.93, p=0.0057, paired *t*-test; *Figure 5a,c*, *Table 1*). We hypothesized that this increase in mean firing rate during locomotion was due to changes in the spike burst properties. We encountered a diversity of rhythmic bursting and non-bursting neurons in the MS. The common features that we recognized for orchid cells were: (1) a high burst incidence during both locomotion and immobility (>3.0 Hz); (2) long duration bursts during locomotion and immobility (median >50 ms); (3) an increase in firing during locomotion (*Figure 5c*); (4) strong coupling mostly around the peak of CA1d theta oscillations (mean vector lengths > 0.2; *Figures 5d–f* and *2f*, *Tables 1* and *3*). We occasionally observed orchid cell bursts spanning two consecutive theta cycles (*Figure 2b*). We did not observe any non-EC projecting MS neurons (e.g. septo-hippocampal neurons) that exhibited all 4 features of the reported orchid cells (data not shown).

We encountered other neurons (n = 7 unlabeled and n = 1 labeled PV+/SATB1+ mGluR1a + neuron) that matched the criteria and were classified as putative orchid cells (*Figure 5d,f,g*, *Figure 5—figure supplement 1b*, *Tables 4*, *8* and *9*). These cells also had different mean firing rates during locomotion and immobility (61.7 ± 18.0 Hz LM versus 43.4 ± 8.9 Hz IM; n = 8, $t_7$ = 3.80, p=0.0067, paired *t*-test; *Table 8*). As with the labeled neurons, most putative orchid cells showed a phase preference close to the peak of CA1d theta oscillations (group mean phase (circular mean ± circular s. d.): identified, 178.6° ± 39.1°, n = 8; putative, 183.3° ± 38.5°, n = 8; difference = 4.7°; p=0.8143, permutation test; *Figure 5f*, *Tables 1* and *9*). Putative orchid cells were also coupled to mid-gamma oscillations, and like orchid cells maintained a peak phase preference (n = 3/8 tested neurons, *Table 9*). To test the reliability of our prediction of orchid cell identity by these signature firing patterns, we recorded then labeled two medial septal neurons that had similar long bursts (TV77q, median (and IQR), 69.0 (48.1) ms; TV78l, 86.3 (60.5) ms; *Table 10*) but a low burst incidence (TV77q, mean ± s.d., 1.5 ± 1.2 Hz; TV78l, 1.8 ± 1.2 Hz; *Figure 5d*). These two additional cells, which were

**Table 3.** Firing patterns of identified orchid cells.

Abbreviations: u, <u>u</u>nknown/<u>u</u>navailable. Spike burst defined as > 3 spikes with ISIs < 40 ms. Intraburst frequency (Hz) is expressed as mean ± s.d.; burst duration and interburst interval (ms) as median and interquartile range. LM, locomotion; SM, small movements including limbs, tail, and shifts in posture; WS, high-frequency whisking and/or sniffing in the absence of other movements; IM, immobility. KS, Kolmogorov-Smirnov.

| Cell name | | TV58g | TV50a | MS14i | MS19b | AJ50j | AJ50h | AJ44j | MS74b |
|---|---|---|---|---|---|---|---|---|---|
| Behavioral states | Burst duration LM | 87.0, 79.8 | 66.8, 30.5 | 121.0, 193.0 | 96.2, 57.5 | 63.2, 22.6 | 70.1, 36.3 | 68.2, 48.9 | 80.1, 40.1 |
| | Burst duration SM | 86.7, 47.5 | 71., 42.0 | 84.2, 56.7 | 85.7, 53.4 | 54.0, 23.8 | 65.2, 56.3 | 70.1, 38.3 | u |
| | Burst duration WS | 91.8, 77.1 | u | 91.7, 61.5 | 92.0, 56.8 | 59.2, 30.2 | 68.3, 35.5 | 52.4, 30.3 | 91.5, 91.9 |
| | Burst duration IM | 94.6, 65.5 | 70.2, 46.0 | 81.1, 48.9 | 100.0, 57.3 | 57.8, 33.3 | 73.9, 33.7 | 68.6, 42.7 | 97.0, 75.0 |
| | Two-sample KS test *P*-value burst duration LM-IM | 0.5928 | 0.0021 | <0.0001 | 0.6573 | 0.0003 | 0.3998 | 0.1947 | 0.0293 |
| | Interburst interval LM | 187.7, 80.9 | 146.0, 35.2 | 179.6, 162.5 | 193.5, 91.8 | 135.8, 41.3 | 160.0, 148.9 | 129.0, 52.1 | 155.3, 86.4 |
| | Interburst interval SM | 214.2, 81.2 | 159.9, 49.3 | 178.7, 99.1 | 230.1, 166.5 | 159.9, 66.1 | 257.6, 209.6 | 158.2, 67.6 | u |
| | Interburst interval WS | 205.1, 100.3 | u | 189.9, 105.1 | 255.4, 180.0 | 178.1, 137.8 | 291.7, 315.5 | 157.3, 73.9 | 254.5, 229.1 |
| | Interburst interval IM | 203.3, 94.5 | 177.3, 92.3 | 180.3, 87.5 | 238.3, 161.8 | 163.3, 66.2 | 255.8, 201.0 | 161.5, 61.7 | 249.0, 135.2 |
| | Two sample KS test *P*-value interburst interval LM-IM | 0.0725 | <0.0001 | 0.0039 | <0.0001 | <0.0001 | <0.0001 | <0.0001 | <0.0001 |
| | Intraburst frequency LM | 95.9 ± 22.1 | 111.4 ± 24.8 | 95.8 ± 24.4 | 83.7 ± 24.6 | 136.1, 27.2 | 75.9 ± 23.6 | 131.4 ± 30.9 | 95.2 ± 20.2 |
| | Intraburst frequency SM | 84.9 ± 19.5 | 105.7 ± 24.5 | 99.7 ± 30.0 | 82.8 ± 27.5 | 121.0, 27.2 | 79.2 ± 17.7 | 96.0 ± 24.5 | u |
| | Intraburst frequency WS | 96.6 ± 24.3 | u | 93.1 ± 24.3 | 80.7 ± 22.7 | 110.6, 27.4 | 71.8 ± 10.2 | 96.4 ± 13.6 | 75.6 ± 16.1 |
| | Intraburst frequency IM | 94.7 ± 22.8 | 113.1 ± 34.0 | 100.6 ± 28.0 | 77.6 ± 19.7 | 124.0, 32.6 | 78.1 ± 16.8 | 89.3 ± 18.2 | 79.3 ± 22.9 |
| | Two sample KS test *P*-value intraburst freq LM-IM | 0.3102 | 0.0693 | 0.0357 | 0.0180 | <0.0001 | 0.0602 | <0.0001 | <0.0001 |
| CA1d mid-gamma (55 – 80 Hz) | Preferred gamma phase | u | u | 164° | 158° | u | 191° | u | u |
| | Mean vector length (*r*) | u | u | 0.04 | 0.25 | u | 0.19 | u | u |
| | Rayleigh *P*-value | u | 0.0860 | 0.0008 | <0.0001 | 0.6225 | <0.0001 | 0.0799 | 0.4700 |
| | Total *n* spikes | u | 1240 | 3553 | 2052 | 1445 | 398 | 579 | 437 |
| CA1d SWRs (130 – 230 Hz) | *n* SWRs | 34 | 40 | 57 | 58 | 0 | 0 | 0 | 16 |
| | Mean rate inside (Hz) | 57.08 | 31.37 | 57.71 | 4.88 | u | u | u | u |
| | Lambda rate outside (Hz) | 44.82 | 49.14 | 48.96 | 33.16 | u | u | u | u |
| | Poisson *P*-value | 0.0181 | 0.0015 | 0.0238 | <0.0001 | u | u | u | u |

DOI: https://doi.org/10.7554/eLife.34395.015

SATB1 +but PV–, projected via the fimbria and innervated the dentate gyrus and CA3 (data not shown). Another group of neurons, the Teevra cells, which target CA3 but not the dentate gyrus, are the most rhythmic neurons of the MSDB (*Joshi et al., 2017*), and thus had a high mean burst incidence (range LM 2.0–7.2 Hz and IM 1.1–5.8 Hz, n = 13 identified cells). However, in contrast to orchid cells, Teevra cells mostly exhibited short duration bursts during locomotion (*Joshi et al., 2017*) and preferentially fired at the *trough* of theta oscillations (group mean phase: identified orchid

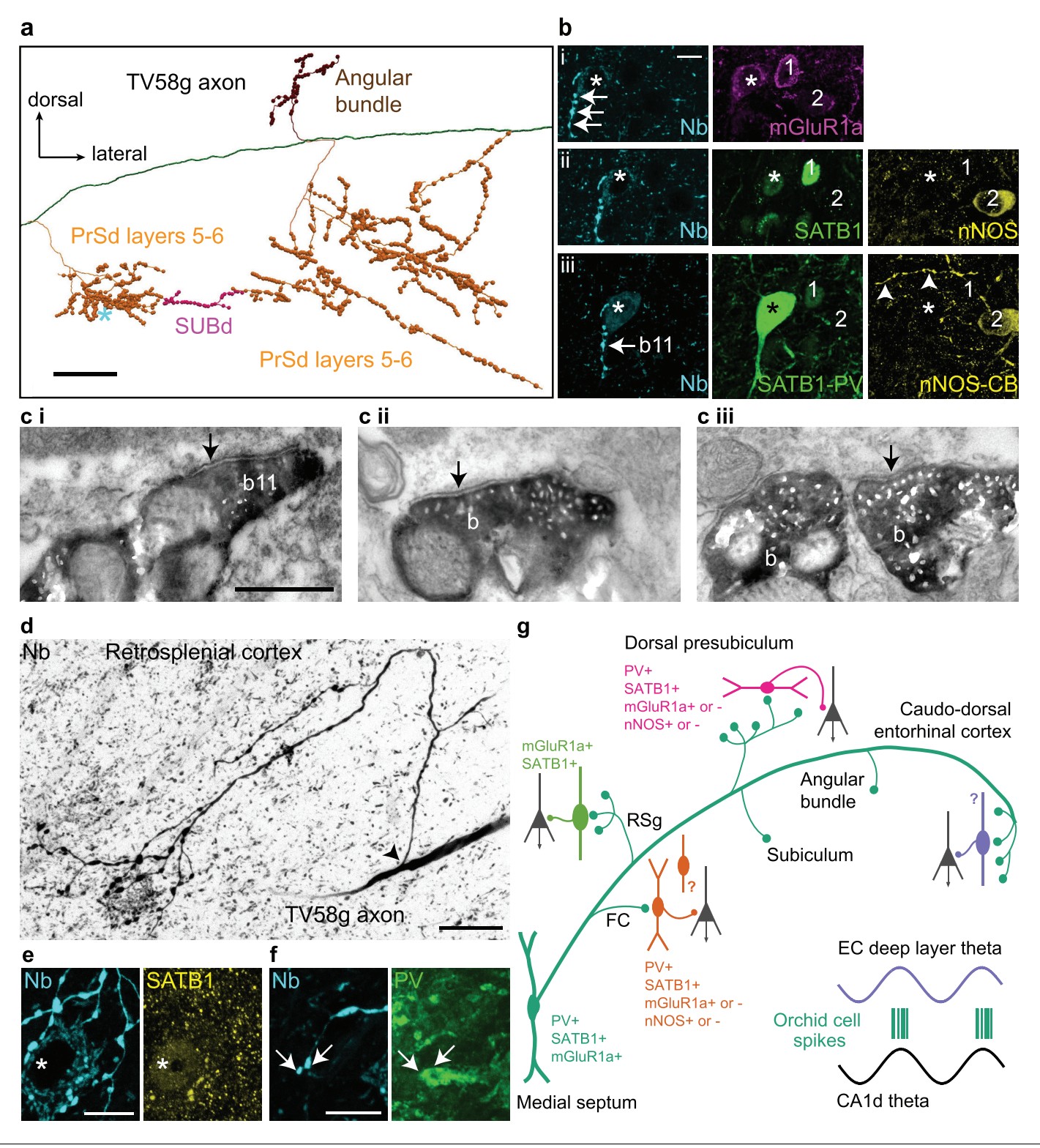

**Figure 4.** Postsynaptic GABAergic target neurons of a GABAergic orchid cell, TV58g. (a) Selected collaterals and varicosities (see *Figure 2a*). Asterisk, location of target neuron in b. (b) Boutons (cyan, neurobiotin, arrows) in the PrSd (asterisk in a) are apposed to a neuron immunopositive for mGluR1a (i, top, magenta, somato-dendritic membrane), SATB1 (ii, middle, green, nucleus), and PV (iii, bottom, green) but lacking detectable immunoreactivity for both nNOS (ii, yellow) and CB (iii, yellow). Arrowheads in iii (bottom right) highlight a CB-immunoreactive process. Two neighboring PV-immunonegative neurons (1, 2) were not targeted. (c) Electron micrographs of synapses (arrows) made by boutons (b) of TV58g on interneurons in the
*Figure 4 continued on next page*

*Figure 4 continued*

PrSd with the proximal dendrite of a PV+ cell (i, b11 in **b**), (ii) the dendrite of an unknown interneuron and (*iii*) soma of another interneuron. (**d**) Main axon and collateral (arrow, branch point) along with terminals surrounding a neuron in layer 6 of RSg (reverse contrast, confocal microscopic image). The target neuron contains a high level of endogenous biotin (high background), revealed by the streptavidin-conjugated fluorophore signal. (**e**) The target neuron soma (asterisk) innervated by multiple terminals (cyan) from (**d**) showed nuclear immunoreactivity for SATB1 (yellow). (**f**) The dendrite of the target showed positive immunoreactivity for PV (green); arrows highlight two terminals in close apposition. Nb, neurobiotin. (**g**) Schematic summary of the data presented in *Figures 3* and *4* and *Tables 1* and *6*. Green circles represent the relative proportions of cortical axon terminals from all orchid cells (see *Table 1*). Immunoreactivity of synaptic target neurons is color-coded based on innervated cortical region. Innervated cortical interneurons may target principal cells (dark gray) on the dendrites, soma or axon initial segment. Bottom right, two idealized theta cycles with representative orchid cell spike bursts at the peak of CA1d theta and EC theta (see *Figure 6*). Images (z-thickness in µm and z-projection type): (**b**) 3.63/4.3/4.76, (*i/ii/iii*), maximum, (**d**) 22.09 standard deviation, (**e**) 2.8 maximum, (**f**) 2.88 maximum. Scale bars: (a) 100 µm, (**b,e,f**) 10 µm, (c) 0.5 µm (applies to *i/ii/iii*), (d) 20 µm.

DOI: https://doi.org/10.7554/eLife.34395.016

cells, 178.6°± 39.1°, n = 8; identified Teevra cells, 37.5°± 53.5°, n = 13; difference = 141.1°; p=0.0388; *Figure 5f*). Overall, no terminals of any of these septo-hippocampal neurons (n = 15 neurons) were observed in extra-hippocampal regions, indicating the reliability of our prediction of orchid cell firing characteristics.

During immobility, with LIA in the hippocampus, orchids cells fired with rhythmic bouts of both long and short bursts, with intermittent non-bursting periods composed of spikes with inter-spike intervals >40 ms (*Figures 5a,e*, *2a,b* and *3g*, *Figure 5—figure supplement 1a,b*). In contrast, once locomotion was initiated, the interval between the bursts changed significantly, revealing the striking shorter rhythmic bursting pattern (group median inter-burst interval (and IQR) during LM = 157.8 (42.7) ms, during IM = 198.1 (69.7) ms, n = 16 neurons, p<0.0001, Sign test; *Figures 5a,g*, *2a,b* and *3f*, *Figure 5—figure supplement 1c*, *Tables 3* and *8*). Median inter-burst intervals were also different between small movements and immobility (SM = 181.8 (61.3) ms, IM = 194.4 (65.9) ms, n = 15 neurons, p=0.0352, Sign test) but not between whisking/sniffing and immobility (WS = 197.9 (73.0) ms, during IM = 201.8 (74.7) ms, n = 13 neurons, p=0.5811, Sign test).

In line with the inter-burst interval, the mean *intraburst frequency* also changed between the behavioral states of locomotion and immobility (109.7 ± 24.6 Hz LM versus 94.8 ± 17.4 Hz IM; n = 16, $t_{15}$ = 3.14, p=0.0067, paired *t*-test). This was reflected by different intraburst frequency distributions during the two behavioral states (n = 12/16 neurons with p<0.05, two-sample Kolmogorov-Smirnov (KS) tests LM versus IM; cell AJ50j but not AJ50h was different during LM versus IM; *Figure 5—figure supplement 1d*, *Tables 3* and *8*), and different distributions of burst durations

**Table 4.** Immunohistochemical profiles of orchid neurons.

Positive (+) or negative (-) immunoreactivity observed within subcellular domain: s, soma; n, nucleus; d, proximal dendrite; a, axon; t, axon terminals; u, unknown (unavailable or inconclusive). *Soma was not recovered; dendrites were beaded indicative of a cell damaged during labeling. **No axon was recovered.

| Cell name | | TV58g | TV50a* | MS14i | MS19b | AJ50h | AJ50j | AJ44j | MS74b* | MS17k** |
|---|---|---|---|---|---|---|---|---|---|---|
| | PV | ad+ | a+ | ds+ | a+ | a+ | a+ | a+ | a+ | ds+ |
| | SATB1 | n+ | u | n+ | n+ | n+ | n+ | n+ | u | n+ |
| | mGluR1a | d+ | u | d+ | s+ | ds+ | ds+ | ds+ | u | ds+ |
| | VGAT | t+ | u | u | u | t+ | u | u | u | u |
| | NK1R | d- | u | u | s- | u | s- | s- | u | ds+ |
| Immunoreactivity | PCP4 | d+ | a+ | ds+ | u | u | u | u | u | u |
| | SMI32 | d+ | u | u | u | u | u | u | u | u |
| | Calbindin | d- | ad- | d- | ad- | u | u | u | a- | d- |
| | Calretinin | d- | ad- | d- | ad- | u | u | u | u | d- |
| | ChAT | sd- | a- | u | u | u | u | u | u | d- |
| | nNOS | sd- | u | u | u | u | u | u | u | u |

DOI: https://doi.org/10.7554/eLife.34395.017

**Table 5.** Marker genes used to define sub-regions of the temporal cortex.

| Marker gene | Gene product | Expression profile | Reference |
|---|---|---|---|
| Amigo2 | Adhesion molecule with Ig like domain 2 | CA2, FC, temporal CA3 | http://mouse.brain-map.org/experiment/show/71250310 |
| Etv1 | Ets variant 1 (ER81) | SUBd, RSg | http://mouse.brain-map.org/experiment/show/72119595 |
| Nos1 | Neuronal nitric oxide synthase 1 | PrS | http://mouse.brain-map.org/experiment/show/75147762 |
| Satb2 | Special AT-rich sequence binding protein 2 | CA1, RSg, PrS | http://mouse.brain-map.org/experiment/show/73992708 |
| Slc17a6 | solute carrier family 17 member 6 (VGLUT2) | SUBd, RSg, PrS | http://mouse.brain-map.org/experiment/show/73818754 |
| Tmem178 | Transmembrane protein 178 | CA2, FC, CA3, EC layer 6 | http://mouse.brain-map.org/experiment/show/73992709 |
| Wfs1 | Wolfram syndrome one homolog (Wolframin) | CA1d, PaS | http://mouse.brain-map.org/experiment/show/74881161 |

DOI: https://doi.org/10.7554/eLife.34395.018

(n = 10/16 cells with p<0.05, two-sample KS tests LM versus IM; *Figure 5e*, *Figure 5—figure supplement 1b*, *Tables 3* and *8*). However, the overall median burst durations did not change between these states (group median burst duration (IQR), LM = 78.5 (20.8) ms, IM = 82.8 (22.9) ms, n = 16 neurons, p=0.2101, Sign test; *Tables 3* and *8*). In summary, the transition from immobility to locomotion results in a shorter inter-burst interval and a higher intraburst frequency, in line with an increase in 5–12 Hz theta power and thus theta-frequency rhythmic bursts.

## Relationship between dorsal hippocampal and caudal entorhinal rhythmic activity

We used a common reference LFP in CA1d for the comparison of spike timing and phase coupling of different orchid cells and other MSDB neurons (*Borhegyi et al., 2004*; *Joshi et al., 2017*). However, orchid cells project to extra-hippocampal areas, which may differ in the temporal dynamics of population activity. Orchid cell terminals were located in the caudo-dorsal EC (*Figure 6a*, *Figure 3—figure supplement 1*), but multi-unit and LFP recordings from rodent EC are typically made in regions of the EC that facilitate their delineation into 'medial' and 'lateral' divisions (*Chrobak and Buzsáki, 1998*; *Igarashi et al., 2014*; *Mizuseki et al., 2009*). In order to compare the relationships of LFPs directly, we targeted the EC and CA1d of mice with glass electrodes and recorded LFPs simultaneously (*Figure 6b–d*). The most accurate way to identify LFP recording sites is by visualizing neurons, hence, we juxtacellularly labeled individual principal cells and/or interneurons at specific depths. *Post hoc* recovery of the labeled neurons revealed that recording sites were in the deep layers of the caudo-dorsal EC in 4 of 8 mice (*Figure 6a–d*). The other four mice had labeled neurons outside the EC; these animals were excluded from analysis. During locomotion, theta oscillations dominated both CA1d and caudo-dorsal EC LFPs, with theta cycles in phase as in the rat (*Mizuseki et al., 2009*) (*Figure 6b,c,e*). Accordingly, EC theta cycle troughs were significantly coupled to CA1d theta cycle troughs (p<0.001 for n = 4 mice (two sites per animal), Rayleigh test; circular mean ±circular s.d., 10.5 ± 15.8°, n = 4 mice (mean of 2 sites per animal); *Figure 6e*). During immobility, irregular LFP dynamics in the two structures showed similar activity, but the EC LFP lacked high-power ripples during hippocampal sharp waves (*Figure 6c*) (*Chrobak and Buzsáki, 1996*). Medial septal neuron firing rate distributions during SWRs were different from a Poisson distribution (n = 4/4 orchid cells from four mice and n = 5 putative orchid cells from four mice; all p<0.05; *Figure 6—figure supplement 1*, *Tables 3* and *9*), with 4/9 neurons reducing their firing rate during SWRs (*Figure 6—figure supplement 1d,g–i*) (*Borhegyi et al., 2004*; *Dragoi et al., 1999*; *Varga et al., 2008*). Moreover, a transient firing rate increase was observed after ripples, which accounted for some of the bursts that were >200 ms in duration (*Figures 2a,b* and *3f*, *Figure 6—figure supplement 1a,b,d,g*). These data suggest that phase coupling to network oscillations by orchid cells, relative to CA1d, is maintained in the most caudal target region of the cells, the caudo-dorsal EC.

## MSDB neurons immunopositive for PV and/or SATB1 and/or mGluR1a

Finally, we estimated the number of MSDB neurons immunoreactive for combinations of PV, SATB1 and mGluR1a by a stereological method (*Figure 7*), to provide a baseline for all studies in mouse. We found that SATB1 could be used to delineate the borders of the entire MSDB (*Figure 7—figure*

**Table 6.** Cortical target neurons of orchid cell TV58g.

Molecular profiles of presumed postsynaptic neurons, based on close apposition of medial septal terminals; +, detectable positive immunoreactivity or signal; –, undetectable immunoreactivity or signal in vicinity of immunopositive signals; u, unknown (unavailable or inconclusive). Parentheses indicate weak immunoreactivity or signal. Endo-biotin, endogenous biotin (detected with streptavidin-conjugated fluorophore) may indicate high metabolic activity due to a high density of mitochondria, such as in 'fast spiking' neurons.

| Target ID | Location | PV | SATB1 | mGluR1a | Endo-biotin | CCK | CB | nNOS | Figure |
|-----------|----------|----|-------|---------|-------------|-----|-----|------|--------|
| S45A | FC | – | + | + | – | – | – | – | |
| S45B | | u | + | u | – | – | u | – | |
| S45D | | + | + | – | + | – | – | + | |
| S45G | | + | + | – | + | – | – | + | |
| S44A | RSg L6 | – | + | + | – | – | – | – | 4d-f |
| S44B | | u | + | u | – | – | u | – | |
| S38A | PrSd L5-6 | + | + | + | + | – | – | – | 4b |
| S38C | | + | + | u | + | – | – | – | |
| S38I | | u | + | + | (+) | – | u | + | |
| S38J | | u | + | u | – | – | + | + | |
| S36A | | + | + | u | + | u | u | u | |
| S34A | | + | + | u | + | u | u | – | |
| S34C | | + | + | – | (+) | u | u | + | |
| Total + | | 7 | 13 | 4 | 8 | 0 | 1 | 5 | |

DOI: https://doi.org/10.7554/eLife.34395.019

supplement 1a–c). We estimated the neuronal population containing at least one of the three molecules as 50,680 ± 2422 neurons (mean ± SEM, from 422 ± 35 counted neurons (mean ± s.d.), n = 3 mice, *Figure 7—figure supplement 1d*). The largest group comprised PV–/SATB1+/mGluR1a– neurons (37.9 ± 2.5%, *Figure 7b*, *Figure 7—figure supplement 1d*), which were also the most numerous cells projecting to the EC (*Figure 1f*). Triple immunopositive neurons, which include the orchid cells, represented 7.2 ± 0.8% of the population. The proportion of PV+/SATB1+/mGluR1a + neurons was similar to PV+/SATB1+/mGluR1a– neurons (7.7 ± 0.5%), but no cell of the latter group was observed to project to the EC (*Figure 7b*, *Figure 7—figure supplement 1d*, *Figure 1f*). Apart from PV+/SATB1+/mGluR1– CA3-projecting Teevra cells (*Joshi et al., 2017*) and PV+/SATB1+/mGluR1a + EC projecting orchid cells, the identities of neurons within the other groups remain to be determined. We conclude that orchid cells, comprising <7% of the entire MSDB population of neurons immunoreactive for at least one of the three molecules, form a subpopulation of GABAergic neurons in the dorsal MS. They provide rhythmic GABAergic input to target interneurons during both locomotion and immobility in distinct extra-hippocampal regions, primarily the PrSd and EC (*Figure 4g*).

## Discussion

A wealth of knowledge on the relationship between theta-rhythmic activity in hippocampus and the MSDB points to a critical role in the temporal coordination of mnemonic processes (*Alonso et al., 1987*; *Borhegyi et al., 2004*; *Dragoi et al., 1999*; *Fuhrmann et al., 2015*; *Gaztelu and Buño, 1982*; *Huh et al., 2010*; *Joshi et al., 2017*; *Kaifosh et al., 2013*; *Leão et al., 2012*; *Mamad et al., 2015*; *McNaughton et al., 2006*; *Mizumori et al., 1989*; *Petsche et al., 1962*; *Vandecasteele et al., 2014*; *Viney et al., 2013*; *Yoder and Pang, 2005*). A large proportion of MSDB projections target extra-hippocampal areas, including the EC (*Fuchs et al., 2016*; *Gonzalez-Sulser et al., 2014*; *Jeffery et al., 1995*; *Justus et al., 2017*; *Unal et al., 2015*). GABAergic MSDB neurons projecting to the *medial* EC preferentially innervate layers II and V (*Gonzalez-Sulser et al., 2014*), and the small PV+ subpopulation of GABAergic MSDB neurons provides a preferential input to 'fast spiking' interneurons in layer II (*Fuchs et al., 2016*). The cellular organization of these circuits may be more diverse than previously anticipated, as we reveal by extracellular recording and selective targeting of

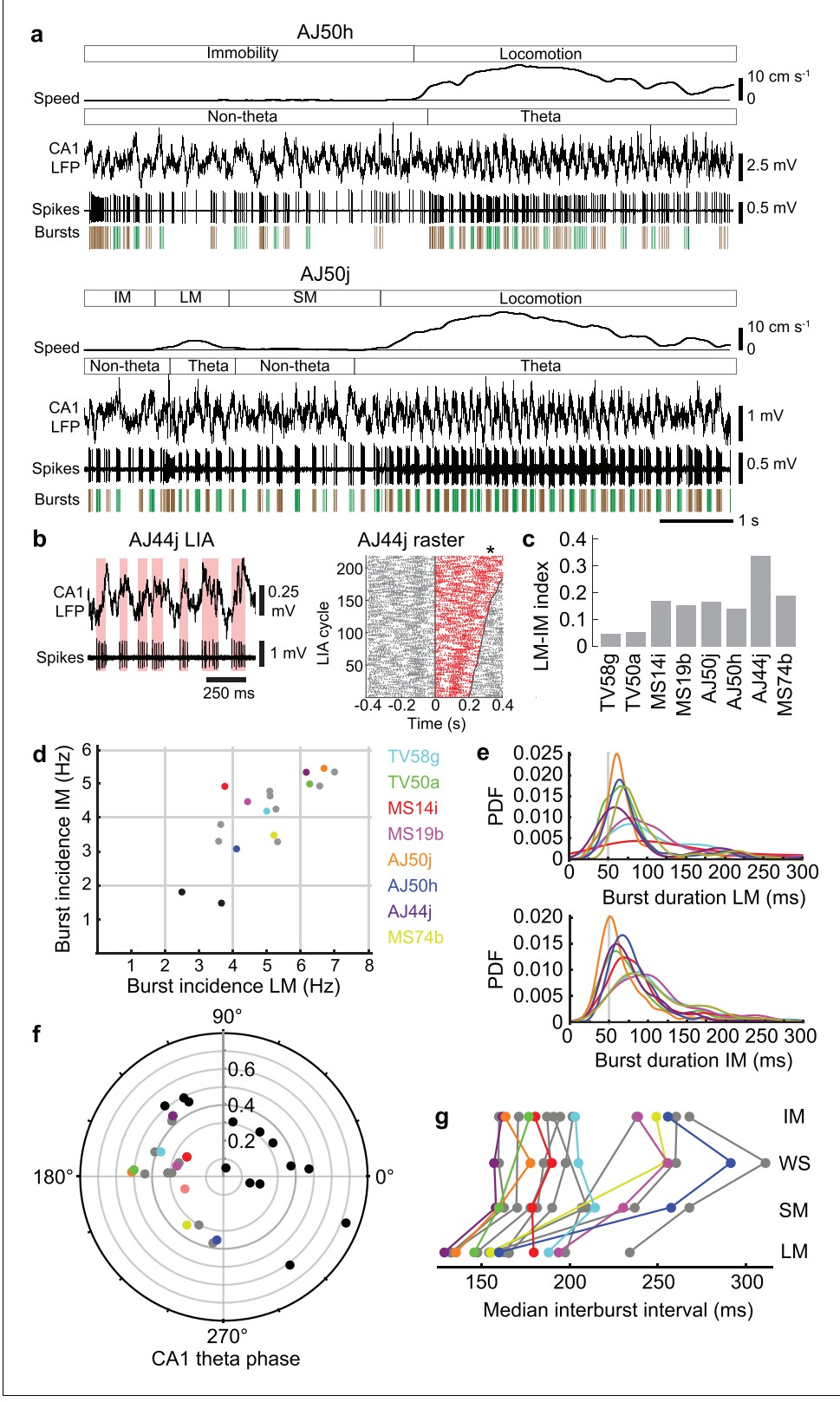

**Figure 5.** Orchid cell firing reflects behavioral states. (**a**) Firing of orchid cells AJ50h (top) and AJ50j (bottom), recorded consecutively in the same session. Algorithm detected bursts are shown below spikes (sequential bursts marked alternately green and brown). Note periods of theta oscillations and non-theta periods during different behavioral states. (**b**) Left, rhythmic burst firing of orchid cell AJ44j during CA1 LIA ('non-theta' periods), with red

*Figure 5 continued on next page*

*Figure 5 continued*

bars highlighting bursts aligned mostly to the ascending/peak phase of LIA oscillations. Right, raster of spikes within (red) and outside (gray) zero-crossings of LIA cycles, ordered by cycle length (minimum interval 200 ms). Time 0 marks the *falling* transition of the LFP (zero crossing). Asterisk, position of the *peak* of the LFP during slower LIA cycles, corresponding to greater spike counts. (c), Mean firing rate index for different states (IM versus LM) for identified orchid cells. All neurons increase their firing from IM to LM. (d) Burst incidence during IM versus LM for identified orchid cells (color-coded, cell IDs on right side), putative orchid cells (gray) and septo-hippocampal cells (black). (e) Probability density functions (PDF) of burst durations of identified orchid cells (color-coded as in c) during LM (top) and IM (bottom), showing that peak burst durations are greater than 50 ms (gray line). (f) Preferred CA1 theta phase (degrees) of identified orchid cells (color-coded as in d), putative orchid cells (gray), and identified Teevra cells (black, cells from *Joshi et al., 2017*); vector length as radial axis (r). (g) Median interburst interval for identified and putative orchid cells during different behavioral states (points; some neurons lack particular states). LM, locomotion; SM, small movements; WS, whisking/sniffing without movement; IM, immobility.

DOI: https://doi.org/10.7554/eLife.34395.020

The following figure supplement is available for figure 5:

**Figure supplement 1.** Burst firing characteristics during different behavioral states.

DOI: https://doi.org/10.7554/eLife.34395.021

---

neurons with particular firing patterns for labeling. Our observation of exclusively extra-hippocampal projecting GABAergic medial septal orchid cells innervating the caudo-dorsal EC reveals a specialized pathway that directly modulates extra-hippocampal circuits while bypassing the hippocampus.

We found the population of PV+/SATB1+/mGluR1a+ neurons within the MSDB to be relatively small, but the viral tracing revealed that a proportion of these innervated the EC, including those within the rostro-dorsal MS. It remains to be determined what other kinds of MSDB neurons apart from GABAergic orchid cells express these three markers. Orchid cells may correspond to the 'constitutively bursting' PV+ neurons with long burst durations and high intraburst frequencies identified in urethane-anesthetized rats (*Borhegyi et al., 2004*; *Simon et al., 2006*) (see also Type 1A cells in [*Alonso et al., 1987*; *Gaztelu and Buño, 1982*; *King et al., 1998*]). Like orchid cells, neurons reported by (*Borhegyi et al., 2004*) had local axon collaterals, which were proposed to inhibit theta-trough firing MSDB neurons, but both the theta-trough and theta-peak firing MSDB neurons were hypothesized to project the hippocampus, in contrast to orchid cells. In awake mice, four major clusters of highly rhythmic medial septal neurons have recently been defined (*Joshi et al., 2017*). The Teevra cell cluster had short burst durations during locomotion and did not significantly change their mean firing rate between locomotion and rest periods, whereas the Komal cell cluster had long burst durations and increased their firing during locomotion compared to rest periods. Juxtacellular labeling of Teevra cells revealed that they targeted CA3 but no extra-hippocampal areas. These CA3-projecting neurons preferentially fired at the trough of CA1d theta oscillations. Komal cells were not labeled but most preferentially fired around the peak of CA1d theta oscillations. In contrast to Teevra cells, rhythmically bursting orchid cells, reported here, increase firing during locomotion, have long duration bursts, fire preferentially at the peak of CA1d theta oscillations, and innervate the PrSd and EC. Thus, orchid cells represent a distinct subpopulation of Komal cells defined by their theta phase firing preferences and synaptic target regions. The other subpopulation of Komal cells that preferentially fire at the theta trough (*Joshi et al., 2017*) remain to be defined using juxtacellular labeling in awake animals, as we have focussed on theta peak firing cells for the very difficult labeling experiments in the present study.

The sites of cortical axon terminations of orchid cells reflect well the major projections arising specifically from the PrSd that terminate in both the RSg and caudo-dorsal EC (*van Groen and Wyss, 1990*; *Vann et al., 2009*). To the best of our knowledge, most in vivo recordings within the EC have been localized to the well-characterized regions of medial and/or lateral EC (*Burgalossi et al., 2014*; *Chrobak and Buzsáki, 1998*; *Deshmukh and Knierim, 2011*; *Deshmukh et al., 2010*; *Hafting et al., 2005*; *Igarashi et al., 2014*; *Ray et al., 2017*), thus the caudo-dorsal EC remains to be defined in terms of its firing pattern repertoire. However, it is expected that any grid cells in this region would have tighter grid cell spacing compared to more ventral regions (*Hafting et al., 2005*). Parts of the lateral EC show weaker theta power in the LFP compared to medial EC in rats, along

**Table 7.** Details of mouse behavior.

LM, locomotion; SM, small postural movements; WS, theta-frequency whisking and/or sniffing without small movements; IM, immobility. *LM periods for these recordings were classified based on high amplitude long-duration (>1 s) signals from the accelerometer with high theta power, and short-duration low amplitude accelerometer signals as SM; those periods outside movement were classed as IM, which may include some whisking/sniffing. Training: limited (limited exposure to head-fixation prior to recording the cell, unfamiliar environment); moderate (typical exposure, familiar with environment); fully trained (fully habituated, familiar with environment and head-fixation). Mouse activity is ordered from the most to least frequent.

| Animal name | Cell ID | Recording conditions | Training | Monitoring | Mouse activity |
|---|---|---|---|---|---|
| TV58 | TV58g | Circular treadmill | Moderate | Video, EMG, accelerometer | IM, WS, SM, LM |
| TV50 | TV50a | Circular treadmill | Moderate | Accelerometer, no video* | IM/(WS), LM, SM |
| MS14 | MS14i | Circular treadmill | Moderate | Video, wheel movement | IM, SM, WS, LM |
| MS19 | MS19b | Circular treadmill | Limited | Video, wheel movement | SM, LM, WS, IM |
| AJ50 | AJ50j | Circular treadmill | Fully trained | Video, wheel movement | IM, WS, SM, LM |
| | AJ50h | Circular treadmill | Fully trained | Video, wheel movement | LM, IM, SM, WS |
| AJ44 | AJ44j | Circular treadmill | Fully trained | Video, wheel movement | IM, SM, LM, WS |
| MS74 | MS74b | Frisbee (with netting) | Fully trained | Video, accelerometer | IM, LM, WS |
| TV77 | TV77q | Circular treadmill | Moderate | Video, wheel movement | IM, WS, SM, LM |
| TV78 | TV78d | Circular treadmill | Moderate | Video, wheel movement | IM/(WS), LM, SM. Whiskers not in focus in video. |
| | TV78k | Circular treadmill | Moderate | Video, wheel movement | IM, SM, LM, WS |
| | TV78l | Circular treadmill | Moderate | Video, wheel movement | IM, SM, (LM), (WS) Whiskers not in focus in video. Slow LM. |
| MS17 | MS17k | Circular treadmill | Moderate | Video, wheel movement | IM, LM, SM, WH |
| MS58 | MS58i | Running disc | Moderate | Accelerometer, no video* | IM/(WS), SM, LM |
| MS24 | MS24a | Circular treadmill | Moderate | Video, wheel movement | IM, LM, WS, SM |
| MS84 | MS84f | Frisbee (with paper) | Fully trained | | IM, LM, SM, WS |
| TV72 | TV72n | Circular treadmill | Limited | Video, wheel movement | IM, SM, WS, LM. 50% of LM was involuntary. |
| TV85 | TV85e | Running disc | Limited | Accelerometer, video | IM, WS, SM, LM. 66% of LM was involuntary. |
| TV86 | TV86b | Running disc | Moderate | Accelerometer, video | IM, WS, LM, SM |

DOI: https://doi.org/10.7554/eLife.34395.022

with lateral EC neurons displaying weaker theta modulation (*Deshmukh et al., 2010*). Based on the medial septal input revealed here, we predict that the caudo-dorsal EC contains interneurons and principal cells with strong theta and gamma modulation (*Beed et al., 2013*). The shared cortical target regions of single orchid cells demonstrate that orchid cells provide strong GABAergic input to presubicular, retrosplenial, and entorhinal interneurons at the peak of CA1d/EC theta cycles during locomotion, and at the peak of slower (<5 Hz) cortical activity during immobility. Such multi-area innervation is likely to support the firing dynamics of spatially-modulated neurons, including grid cells, which lose their spatial periodicity upon MS inactivation (*Brandon et al., 2013*). Encoding of information takes place preferentially at the theta peak (*Manns et al., 2007*), coincident with the preferential firing of layer 3 EC principal neurons that project to CA1d (*Mizuseki et al., 2009*). Therefore, disinhibition of EC glutamatergic neurons by orchid cells around the theta peak may facilitate the transfer of temporal sequences relating to navigation to the hippocampus (*Koenig et al., 2011*; *Schlesiger et al., 2015*).

Principal cell assemblies are synchronized over a 25 ms time scale (*Harris et al., 2003*), which matches the membrane time constant of hippocampal pyramidal cells (*Spruston and Johnston, 1992*) and is within the range of gamma oscillations. Orchid cells were able to couple their firing to CA1d mid-gamma frequency oscillations, which originate in the EC (*Colgin et al., 2009*; *Lasztóczi and Klausberger, 2016*; *Schomburg et al., 2014*) where orchid cell terminals are located. One potential source of gamma and ripple modulation in the MS is directly from GABAergic hippocampo-septal cells (*Jinno et al., 2007*; *TothTóth et al., 1993*). The degree of coupling by individual orchid cells to mid-gamma oscillations (from lack of coupling to strong coupling) may depend on the

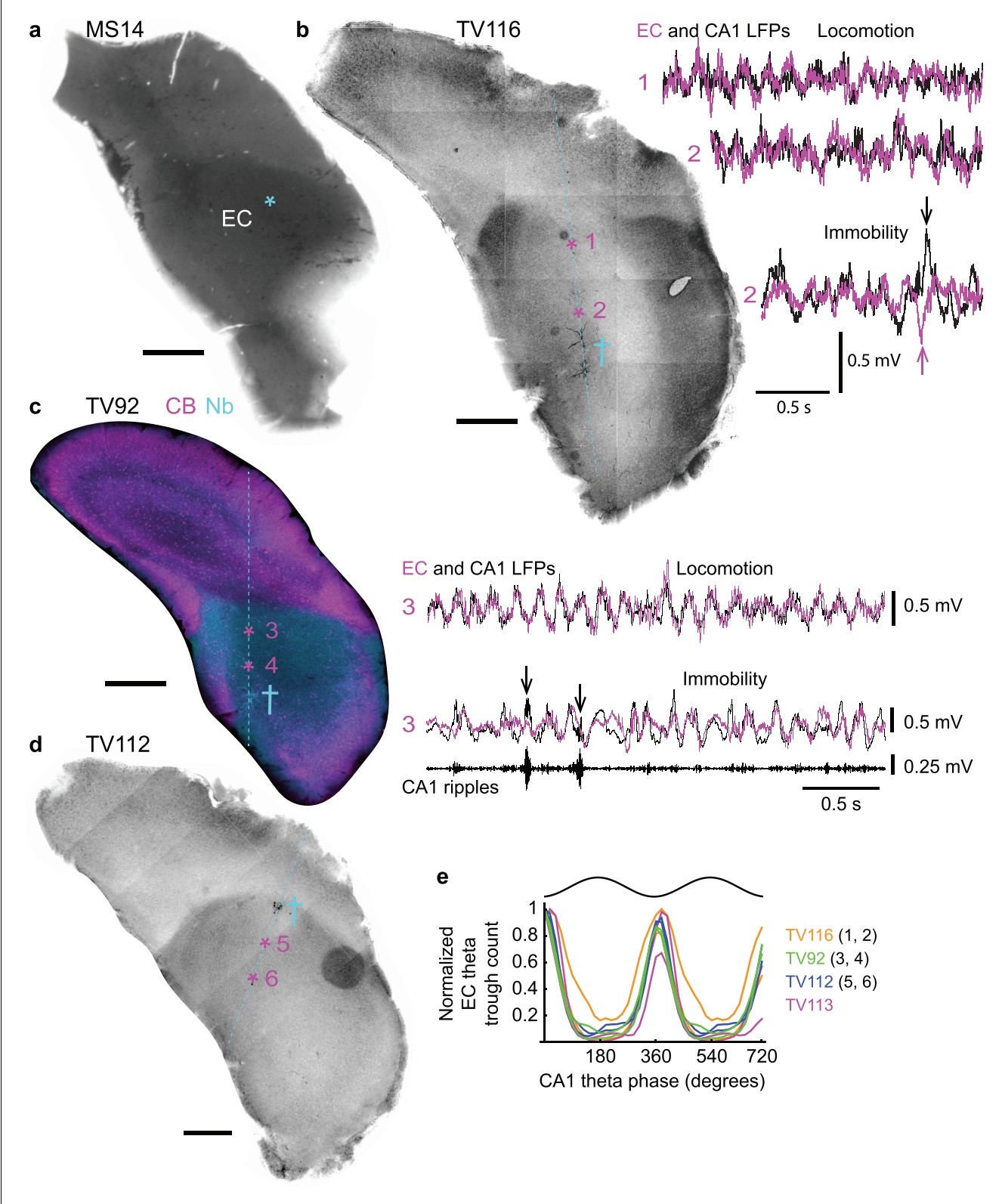

**Figure 6.** Phase relationship of LFPs in dorsal CA1 and entorhinal cortex. (**a**) The position of axon terminals from orchid cell MS14i in the caudo-dorsal EC is marked by an asterisk (DAB-based HRP reacted, 70 μm-thick coronal section flipped horizontally). (**b–d**) Coronal sections from different animals (TV116, TV92, TV112) showing the EC at similar rostro-caudal levels to (**a**) with position of glass electrode tracts (cyan, dashed lines). Cross symbols mark sites of labeled neurons used to assess relative depth of LFP recording sites. (**b**) Left, reverse contrast fluorescence image (neurobiotin signal).

*Figure 6 continued on next page*

*Figure 6 continued*

Asterisks mark two LFP recording sites. Right, overlaid entorhinal (magenta) and CA1d (black) LFPs recorded at sites 1 and 2. Arrows, sharp waves. (c) Immunoreactivity for CB (magenta) and visualization of neurobiotin (Nb, cyan; widefield epifluorescence). Asterisks mark two LFP recording sites. Right, LFP recordings from site three during locomotion and immobility. Ripples, 130–230 Hz filtered CA1d LFP (also marked by arrows in the LFP). (e) Theta phase histogram for EC LFP theta (5–12 Hz) trough counts (normalized) relative to CA1d theta phase (n = 4 mice, two sites per mouse). Scale bars for all images: 500 μm.

DOI: https://doi.org/10.7554/eLife.34395.026

The following figure supplement is available for figure 6:

**Figure supplement 1.** Ripple-related activity of orchid cells and putative orchid cells.

DOI: https://doi.org/10.7554/eLife.34395.027

level of habituation/training of the animal. During each theta cycle, orchid cells may coordinate the firing of principal cell assemblies in 'upstream' (PrSd) and 'downstream' (RSg, EC) regions via the phasic modulation of innervated GABAergic interneurons. We present two main hypotheses. (1) During the peak of the CA1d theta cycle, theta-peak firing orchid cells could inhibit target interneurons that preferentially fire during the trough of theta, leading to disinhibition of principal cells around the theta peak, including EC principal neurons projecting to CA1. This is counter-phased by the theta-trough firing Teevra cells and their theta-peak firing CA3 axo-axonic cell target interneurons (*Joshi et al., 2017*; *Viney et al., 2013*), leading to disinhibition of CA3 pyramidal neurons on the descending phase (*Mizuseki et al., 2009*), which project to CA1. (2) Alternatively or in addition, orchid cells could target theta-peak firing cortical interneurons that are modulated at mid-gamma frequency, leading to mid-gamma modulation of PrSd and EC principal cells around the peak of theta cycles (*Manns et al., 2007*). This extra-hippocampal pathway may contribute to the synchronization of inter-areal cortico-cortical loops via theta-nested gamma frequency rhythmic inhibition of cortical GABAergic circuits (*Jeffery et al., 1995*; *Kim et al., 2015*). Such a population rhythm could then be relayed to the hippocampus from the EC (*Chrobak et al., 2000*; *Igarashi et al., 2014*; *Mizuseki et al., 2009*). A similar temporal coordination with a wider temporal window may exist during quiescent periods, when theta power is lower and the intraburst frequency of orchid cells decreases.

Orchid cells may also coordinate SWRs. Ripples also occur downstream of CA1 within deep layers of the PrS and EC (*Chrobak and Buzsáki, 1996*), the target regions of orchid cells. Since ipsilateral entorhinal ripples occur 5–30 ms after CA1d ripples (*Chrobak and Buzsáki, 1996*), it is possible that the increase in firing of orchid cells following CA1d SWRs coordinates the offline replay/recall of spatial memory traces within extra-hippocampal areas; most SWRs were probably 'awake' SWRs under our conditions (*Roumis and Frank, 2015*). Different levels of arousal may account for differences in firing rates during SWRs.

Cortical target interneurons of orchid cells included PV+ and nNOS+ neurons. Most cortical PV+ neurons are considered to be 'fast spiking' basket and axo-axonic cells, and display rhythmic modulation at theta frequency (*Preston-Ferrer et al., 2016*). In the PrSd, some PV+ neurons are immunopositive for nNOS, which is also expressed by other interneuronal types. The in vivo firing patterns of interneurons in FC, PrSd, and RSg are unknown (*Henriksen et al., 2010*) except for 'theta' cells recorded in the vicinity of PrSd head direction cells (*Preston-Ferrer et al., 2016*; *Taube et al., 1990*). 'Fast spiking' neurons have also been recorded in more caudal regions of PrS but do not show strong theta modulation (*Tukker et al., 2015*). Visualization of interneuron axon terminals is required to define their target cortical layers as well as their target subcellular domains on postsynaptic principal cells. Based on the high density of GABAergic MSDB terminals around somata and dendrites of PV+ cells in PrSd, we hypothesize that postsynaptic targets of orchid cells will be strongly theta modulated, supporting the rhythmic coordination of pyramidal cell assemblies. The occasional bursts spanning two theta cycles may be responsible for the 'theta cycle skipping' observed in some EC interneurons (*Brandon et al., 2013*).

In summary, orchid cells provide a specialized source of subcortical GABAergic input to specific extra-hippocampal regions that are functionally-related and converge to the caudo-dorsal EC. Their movement-related theta-rhythmic burst firing, and similarly strong rhythmic bursting during immobility, likely serve as a mechanism - via target interneurons - for the temporal coordination of cortico-cortical neurons. These cortico-cortical principal neurons project from 'upstream' areas (PrSd) to

**Table 8.** Behavior-dependent firing patterns of putative orchid cells.

*Labeled soma observed in the dorsal MS; only a faint projection axon was recovered. **Destroyed soma observed in the dorsal MS; no axon was recovered. Abbreviations: u, u̲nknown/u̲navailable. Spike burst defined as > 3 spikes with ISIs < 40 ms. Firing rates and burst incidence (Hz) are expressed as mean of 1 s bins ± s.d.; intraburst frequency (Hz) as mean ± s.d.; burst duration and interburst interval (ms) as median and interquartile range. LM, locomotion; SM, small movements including limbs, tail, and shifts in posture; WS, high-frequency whisking and/or sniffing in the absence of other movements; IM, immobility. KS, Kolmogorov-Smirnov.

| Cell name | MS17k* | MS58i* | MS24a | MS84f | TV78d | TV78k | TV85e | TV86b |
|---|---|---|---|---|---|---|---|---|
| Max speed (cm/s) | 9.9 | u | 16.8 | u | 14.3 | 4.5 | u | u |
| Firing rate LM | 51.6 ± 25.5 | 80.9 ± 20.2 | 48.7 ± 10.7 | 65.2 ± 7.5 | 31.5 ± 11.9 | 60.6 ± 12.5 | 87.6 ± 14.9 | 67.9 ± 9.1 |
| Firing rate SM | 51.8 ± 13.8 | 42.3 ± 9.1 | 50.1 ± 11.7 | 56.0 ± 10.9 | 41.6 ± 18.3 | 34.4 ± 12.1 | 47.3 ± 14.3 | 49.5 ± 6.4 |
| Firing rate WS | 46.7 ± 13.0 | 52.1 ± 14.7 | 44.2 ± 11.5 | 48.9 ± 11.6 | u | 30.3 ± 15.5 | 50.1 ± 10.1 | 55.1 ± 9.3 |
| Firing rate IM | 44.6 ± 13.8 | 52.1 ± 14.7 | 33.7 ± 8.8 | 48.4 ± 9.2 | 39.5 ± 9.8 | 33.6 ± 9.6 | 58.0 ± 13.0 | 37.0 ± 9.7 |
| Burst incidence LM | 3.3 ± 2.2 | 6.6 ± 1.0 | 5.3 ± 1.4 | 5.1 ± 1.5 | 3.6 ± 1.3 | 3.6 ± 1.0 | 7.0 ± 1.4 | 5.3 ± 1.2 |
| Burst incidence SM | 4.7 ± 1.0 | 4.9 ± 1.4 | 4.0 ± 0.9 | 4.6 ± 1.4 | 4.0 ± 1.1 | 3.3 ± 0.9 | 5.2 ± 1.5 | 5.0 ± 0.0 |
| Burst incidence WS | 4.7 ± 1.3 | u | 3.4 ± 1.0 | 4.8 ± 1.3 | u | 2.8 ± 1.0 | 5.4 ± 1.1 | |
| Burst incidence IM | 4.3 ± 1.2 | 4.9 ± 1.1 | 3.3 ± 1.2 | 4.8 ± 1.1 | 3.8 ± 1.0 | 3.3 ± 1.0 | 5.3 ± 1.1 | |
| Burst duration LM | 86.7, 115.8 | 70.2, 22.0 | 57.4, 36.8 | 86.4, 111.8 | 76.9, 39.4 | 156.3, 101.6 | 64.1, 34.0 | 89.6, 62.7 |
| Burst duration SM | 90.8, 84.8 | 67.3, 47.2 | 107.0, 84.0 | 99.2, 77.6 | 90.0, 57.6 | 112.9, 70.2 | 62.5, 25.9 | 101.2, 77.0 |
| Burst duration WS | 82.9, 74.0 | 76.7, 57.4 | 97.3, 65.1 | 96.7, 93.0 | u | 118.3, 50.0 | 62.1, 43.4 | 96.5, 70.3 |
| Burst duration IM | 84.4, 67.7 | 76.7, 57.4 | 94.1, 73.0 | 94.3, 69.9 | 95.3, 72.1 | 112.6, 72.3 | 65.9, 45.1 | 75.8, 43.4 |
| Two sample KS test *P*-value burst duration LM-IM | 0.3902 | 0.0024 | <0.0001 | 0.0294 | <0.0001 | <0.0001 | 0.5547 | 0.0045 |
| Interburst interval LM | 166.6, 155.2 | 133.1, 31.1 | 155.6, 96.5 | 147.4, 115.9 | 197.8, 163.9 | 233.8, 155.6 | 133.0, 32.1 | 163.1, 91.8 |
| Interburst interval SM | 190.7, 135.2 | 169.8, 117.3 | 236.9, 138.7 | 178.1, 86.8 | 206.6, 136.8 | 267.4, 163.5 | 163.1, 73.2 | 181.8, 84.8 |
| Interburst interval WS | 197.9, 95.7 | u | 259.9, 194.5 | 189.1, 92.9 | u | 311.2, 299.0 | 159.7, 79.0 | 185.4, 95.3 |
| Interburst interval IM | 201.8, 117.9 | 170.7, 100.1 | 260.7, 186.0 | 187.2, 88.1 | 238.1, 136.9 | 267.6, 137.6 | 159.9, 91.8 | 194.4, 113.2 |
| Two sample KS test *P*-value inter-burst interval LM-IM | 0.0484 | <0.0001 | <0.0001 | <0.0001 | <0.0001 | 0.0130 | <0.0001 | <0.0001 |
| Intraburst frequency LM | 103.4 ± 35.6 | 141.9, 24.7 | 144.3, 46.8 | 105.1, 23.8 | 83.3, 22.9 | 81.8, 17.9 | 150.2, 32.8 | 119.4, 32.4 |
| Intraburst frequency SM | 94.2 ± 28.5 | 101.9, 26.1 | 104.4, 35.8 | 89.3, 27.0 | 87.4, 22.8 | 68.0, 14.6 | 117.9, 28.2 | 96.3, 24.6 |
| Intraburst frequency WS | 95.6 ± 38.8 | u | 97.8, 25.9 | 88.9, 24.2 | u | 70.8, 17.8 | 118.2, 26.9 | 105.0, 26.8 |
| Intraburst frequency IM | 95.0 ± 27.3 | 101.9, 30.3 | 82.5, 27.4 | 86.9, 22.7 | 89.4, 27.2 | 72.4, 19.8 | 135.5, 39.0 | 96.3, 25.4 |
| Two sample KS test *P*-value intraburst frequency LM-IM | 0.0914 | <0.0001 | <0.0001 | <0.0001 | 0.0449 | <0.0001 | 0.0003 | <0.0001 |

DOI: https://doi.org/10.7554/eLife.34395.023

'downstream' areas (RSg, EC) forming transient, gamma-timescale assemblies involved in mnemonic processes representing sequential information, such as routes of navigation.

## Materials and methods

### Surgical procedures

All procedures involving experimental animals were approved by the Department of Pharmacology Animal Welfare and Ethical Review Body under approved personal and project licenses in accordance with the Animals (Scientific Procedures) Act, 1986 (UK) and associated regulations. Adult male C57Bl7/J mice (originating from Charles River Laboratories; n = 120 mice for head-plate implantation, 23–42.5 g; n = 10 mice for viral tracing, 22–28 g) were maintained on a 12/12 hr light-dark cycle (lights on during the day), and prior to surgery housed in groups of up to four within individually ventilated cages. Mice were anesthetized with isoflurane (IsoFlo, Abbott) followed by a sub-cutaneous injection of opioid analgesic buprenorphine (Vetergesic, 0.1 mg/kg) and maintained with 1–3% (vol/

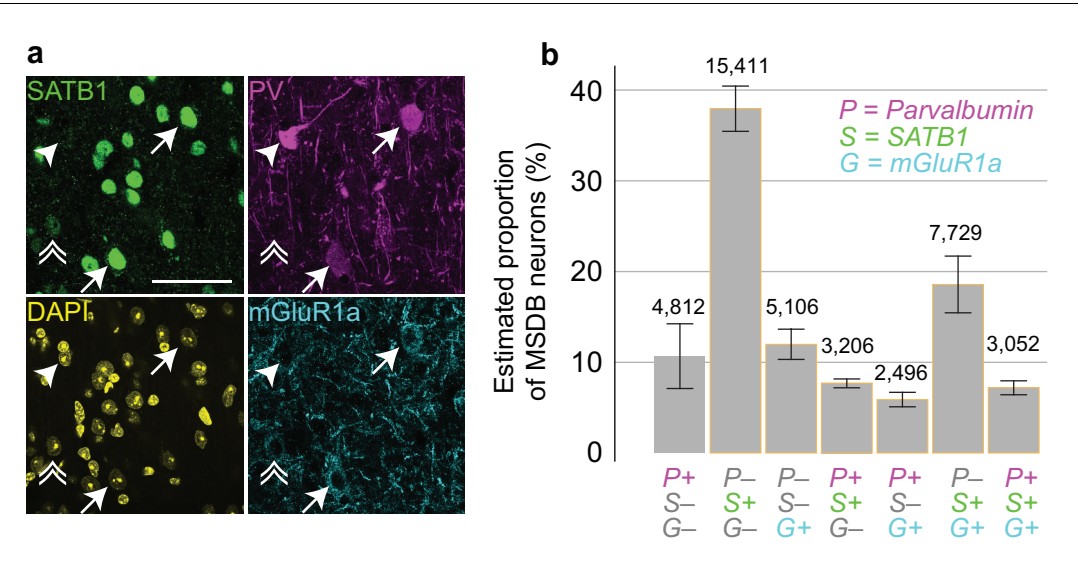

**Figure 7.** Proportions of MSDB neurons immunoreactive for PV, SATB1 and mGluR1a. (a) Confocal microscopic z-projection (3 μm thick) of SATB1 (green), PV (magenta) and mGluR1a (cyan) immunoreactivity in the MS. Arrows, two PV+/SATB1+/mGluR1a+ neurons. Arrowhead, PV+/SATB1–/mGluR1a– neuron. Double arrowhead, PV–/SATB1+/mGluR1+ neuron. DAPI, yellow. Scale bar, 50 μm. (b) Stereological estimation of neuronal subpopulations in the MSDB immunoreactive for combinations of PV (P), SATB1 (S) and mGluR1a (G). Total estimated counts are shown above each bar. Error bars, standard error.

DOI: https://doi.org/10.7554/eLife.34395.028

The following figure supplement is available for figure 7:

**Figure supplement 1.** Stereological estimation of the number of neurons immunoreactive for PV and/or SATB1 and/or mGluR1a across the entire MSDB.

DOI: https://doi.org/10.7554/eLife.34395.029

---

vol) isoflurane. The scalp was clipped and mice were fixed to a stereotaxic frame (Kopf Instruments) using ear bars and a jaw bar. Ocular lubricant was applied, and small volumes of the non-steroidal anti-inflammatory analgesic meloxicam (Metacam, Boehringer Ingelheim) were injected into the scalp. Under aseptic conditions, an incision was made along the scalp at the midline and the skull was exposed.

## Head-plate implantation

Two small M1 screws were placed in holes drilled above the cerebellum, one of which was used as the electrical reference and ground. Another screw was fixed at 1.50 mm anterior and 1.70 mm lateral of Bregma for a frontal cortical EEG; occasionally a second EEG screw was placed over the retrosplenial cortex or V1. A machined glass reinforced plastic head-plate (either a 0.7 g or 1.1 g version, custom made at the Department of Physics, Oxford University) was positioned over the screws, and bone cement was used to fix the head-plate and screws to the skull. Craniotomies were made above the MSDB (0.85 mm anterior and 0 mm lateral of Bregma) and right CA1d (2.50 mm posterior and 1.70 mm lateral of Bregma). In 8/120 mice, a craniotomy was made above the right EC (4.85 mm posterior and 3.10 mm lateral of Bregma for seven mice; 4.30 mm posterior and 3.10 mm lateral for one mouse (TV92)). Craniotomy sites were covered in silicon and mice were left to recover (typically 1 d). A standardized distress scoring system was used to monitor the recovery. For some experiments, craniotomies were instead carried out during a second surgery, using the same anesthesia regime as above.

## Viral tracing

Small craniotomies were made above the MS (0.85 mm anterior and 0.60 mm lateral of Bregma, 10° latero-medial angle; 3.65 mm ventral to dura mater) to inject the anterograde Cre-dependent

**Table 9.** Network state-dependent firing patterns of putative orchid cells.
*Labeled soma observed in the dorsal MS; only a faint projection axon was recovered. **Destroyed soma observed in the dorsal MS; no axon was recovered. Abbreviations: u, unknown/unavailable.

| Cell name | | MS17k* | MS58i** | MS24a | MS84f | TV78d | TV78k | TV85e | TV86b |
|---|---|---|---|---|---|---|---|---|---|
| CA1d theta (5 – 12 Hz) | LFP measurement | SP | SO/SP | SO/SP | Alveus | SP | SP | SP | SP |
| | Pref. theta phase | 176° | 133° | 178° | 177° | 261° | 163° | 160° | 243° |
| | Mean vector length (r) | 0.29 | 0.42 | 0.43 | 0.31 | 0.38 | 0.26 | 0.41 | 0.30 |
| | Rayleigh P-value | <0.0001 | <0.0001 | <0.0001 | <0.0001 | <0.0001 | <0.0001 | <0.0001 | <0.0001 |
| | Total n spikes | 727 | 2869 | 3287 | 4624 | 2154 | 1291 | 2792 | 1292 |
| | Spikes per cycle (mean ± s.d.) | 9.0 ± 3.7 | 7.6 ± 3.9 | 6.8 ± 3.9 | 8.8 ± 2.7 | 5.3 ± 3.4 | 7.5 ± 4.1 | 7.3 ± 4.4 | 9.6 ± 3.6 |
| CA1d mid-gamma (55 – 80 Hz) | Pref. gamma phase | u | u | 155° | u | 144° | u | 207° | u |
| | Mean vector length (r) | u | u | 0.3210 | u | 0.12 | u | 0.04 | u |
| | Rayleigh P-value | 0.1919 | 0.6089 | <0.0001 | 0.0870 | <0.0001 | 0.3506 | 0.0417 | 0.4627 |
| | Total n spikes | 1003 | 1032 | 284 | 1227 | 871 | 1457 | 2000 | 1274 |
| CA1d SWRs (130 – 230 Hz) | n SWRs | 24 | 6 | 55 | 0 | 24 | 46 | 19 | 42 |
| | Mean rate inside (Hz) | 69.35 | u | 27.47 | u | 3.75 | 19.67 | u | 4.48 |
| | Lambda rate outside (Hz) | 46.33 | u | 36.92 | u | 40.71 | 31.92 | u | 42.31 |
| | Poisson P-value | 0.0004 | u | 0.0176 | u | <0.0001 | 0.0068 | u | <0.0001 |

DOI: https://doi.org/10.7554/eLife.34395.024

adeno-associated virus (pAAV$_2$-EF1a-DIO-hChR2(H134R)-EYFP; Vector BioLabs; n = 10/10 mice), and above the caudal EC (4.85 mm posterior and 3.20 mm lateral of Bregma; 1.20–1.90 mm ventral to dura mater) to inject mutant pseudorabies virus PRV-hSyn-Cre (PRV IE180-null-hSyn-Cre; gift from Prof Lynn Enquist, Princeton University; CNNV grant no. P40RR018604; (*Oyibo et al., 2014*)) either unilaterally (n = 5/10 mice; animals MS60, MS61, TV89, TV96, TV102) or bilaterally (n = 5/10 mice; MS66, MS76, MS77, MS78, TV95). Membrane-bound expression of ChR2-EYFP from the AAV is dependent on the expression of Cre from PRV-hSyn-Cre within the same neurons. Both viruses were pressure-injected using glass pipettes (tip diameter: 12–20 µm) attached to 1 µl syringes at a rate of ~50 nl/min. Accordingly, we injected 400 nl of AAV into the MSDB (*Unal et al., 2015*), and 200–400 nl of PRV-hSyn-Cre into the caudo-dorsal EC (200 nl/site in MS60, TV89; 250 nl/site in MS61, TV95; 300 nl/site in TV96, TV102; 400 nl/site in MS66, MS76, MS77, MS78). Mice were perfuse-fixed 14–30 days after injections in order to ensure axonal transport of both viruses. Following sectioning, weak EYFP expression was observed within the cytoplasm of a minority of somata at the injection site in two animals (data not shown); strong thick and thin axons could be followed rostrally towards the MSDB and did not originate from the weakly-labeled local EC neurons. Four injected mice were excluded due to lack of EYFP expression. Injection sites were localized based on the lesion caused at the tip of the injection pipette (due to the pressure of the injection). These small lesions, which were often surrounded by a few small patches of accumulated fluorophores, were visible under high magnification and represented the centre of the injection site (*Figure 1—figure supplement 1b*).

### *In vivo* extracellular recordings and juxtacellular labeling

Experiments were carried out in a dedicated recording room during the light phase, typically 1–2 d after the craniotomies. Mice were habituated to a circular treadmill (see *speed measurements* below), a running disc (Fast Trac, LBS Ltd, Surrey, UK), or a Frisbee (radius 15 cm) (*Table 7*), below a stereotaxic frame and attached to a head-restraint device (custom made at the Department of Physics, Oxford University) for short durations (few minutes) then for longer durations. Animals AJ44 and AJ50 were trained to run on the circular treadmill for 7–10 d before the craniotomy. Two separate glass electrodes filled with 2.5–3.0% neurobiotin (wt/vol) in 0.5 M NaCl (10–18 MΩ) were advanced into the brain with micromanipulators (IVM-1000, Scientifica Ltd, Maidenhead, UK), targeting CA1d at a 10° postero-anterior angle (sometimes filled only with 0.5 M NaCl), and the midline dorsal MS

**Table 10.** Firing patterns of identified septo-hippocampal neurons.

(Related to *Figure 5*) Abbreviations: u, <u>u</u>nknown/<u>u</u>navailable. Spike burst defined as > 3 spikes with ISIs < 40 ms. Firing rates and burst incidence are expressed as mean of 1 s bins ± s.d.; intraburst frequency as mean ± s.d.; burst duration and interburst interval as median and interquartile range. LM, locomotion; SM, small movements including limbs, tail, and shifts in posture; WS, high-frequency whisking and/or sniffing in the absence of other movements; IM, immobility.

| Cell name | | TV77q | TV78l |
|---|---|---|---|
| Behavioral states | Firing rate LM (Hz) | 38.4 ± 12.2 | 25.3 ± 1.7 |
| | Firing rate SM (Hz) | 22.5 ± 7.6 | 16.7 ± 5.8 |
| | Firing rate WS (Hz) | 16.0 ± 7.5 | 23.3 ± 8.6 |
| | Firing rate IM (Hz) | 15.6 ± 8.1 | 21.6 ± 6.2 |
| | Burst incidence LM (Hz) | 3.7 ± 1.4 | 2.5 ± 0.6 |
| | Burst incidence SM (Hz) | 2.6 ± 0.9 | 1.5 ± 1.2 |
| | Burst incidence WS (Hz) | 1.5 ± 1.1 | 1.3 ± 1.2 |
| | Burst incidence IM (Hz) | 1.5 ± 1.2 | 1.8 ± 1.2 |
| | Burst duration LM (ms) | 69.7, 73.6 | 85.9, 96.2 |
| | Burst duration SM (ms) | 75.9, 40.5 | 64.9, 19.6 |
| | Burst duration WS (ms) | 74.3, 38.6 | 109.3, 102.4 |
| | Burst duration IM (ms) | 69.0, 48.1 | 86.3, 60.5 |
| | Interburst interval LM (ms) | 196.6, 206.7 | 537.7, 412.8 |
| | Interburst interval SM (ms) | 332.0, 250.9 | 457.3, 418.8 |
| | Interburst interval WS (ms) | 427.4, 588.3 | 427.3, 214.8 |
| | Interburst interval IM (ms) | 445.5, 615.9 | 393.4, 384.6 |
| | Intraburst frequency LM (Hz) | 85.3 ± 27.1 | 69.8 ± 35.9 |
| | Intraburst frequency SM (Hz) | 78.1 ± 28.3 | 65.6 ± 15.9 |
| | Intraburst frequency WS (Hz) | 82.9 ± 27.2 | 69.3 ± 13.3 |
| | Intraburst frequency IM (Hz) | 85.7 ± 31.1 | 64.1 ± 15.6 |
| CA1d theta (5 – 12 Hz) | LFP measurement | SP | SP |
| | Preferred theta phase | 230° | u |
| | Mean vector length (*r*) | 0.27 | u |
| | Rayleigh *P*-value | <0.0001 | 0.2819 |
| | Total *n* spikes | 1206 | 505 |
| | Spikes per cycle (mean ± s.d.) | 3.1 ± 2.7 | u |
| CA1d SWRs(130 – 230 Hz) | *n* SWRs | 50 | 61 |
| | Mean rate inside (Hz) | 6.20 | 0.65 |
| | Lambda rate outside (Hz) | 16.05 | 21.73 |
| | Poisson *P*-value | 0.0025 | <0.0001 |

DOI: https://doi.org/10.7554/eLife.34395.025

(0° angle, near or directly through the sagittal sinus). For EC recordings, a 0° angle was used in 7 of 8 mice. In one mouse (TV92), a 10° anterior to posterior angle was used; this mouse also had a craniotomy over the MS. All signals were amplified x1,000 (ELC-01MX, BF-48DGX and DPA-2FS modules, NPI Electronic). Both wideband (0.3 Hz to 10 kHz) and band-pass filtered (action potentials, 0.8–5 kHz; LFPs, 0.3–500 Hz) signals were acquired in parallel and digitized at 20 kHz (Power1401, Cambridge Electronic Design). HumBugs (Digitimer) were used to remove 50 Hz noise. A video camera was used to monitor behavior, and wheel movement and speed were recorded using an Arduino (see below and *Table 7*). In some experiments, an accelerometer was placed on the wheel to detect wheel movement, and in one experiment (animal TV58) an electromyogram (EMG) was used to help detect movement of the animal (from the neck muscle). Data were recorded in Spike2 software

**Table 11.** Specificity information for primary antibodies.
Rb, rabbit; Gt, goat; Ms, mouse; Gp, guinea pig; Ck, chicken.

| Molecule | Host | Dilution | Source | Specificity information | RRID |
|---|---|---|---|---|---|
| Calbindin (CB) | Rb | 1:5000 | Swant, CB-38 (lot 5.5) | Supplementary Table 2 in (*Viney et al., 2013*) | RRID: AB_2721225 |
| Calretinin (CR) | Rb | 1:500-1:1000 | Swant, 7699/3 hr (lot 18299) | Supplementary Table 2 in (*Viney et al., 2013*) | RRID: AB_10000321 |
| Calretinin (CR) | Gt | 1:1000 | Swant, CG1 | *Table 1* in (*Unal et al., 2015*) | RRID: AB_10000342 |
| Choline acetyltransferase (ChAT) | Gt | 1:500 | Chemicon, AB144P | *Table 1* in (*Unal et al., 2015*) | RRID: AB_2079751 |
| GFP | Ck | 1:500 | Aves Labs, GFP-1020 | *Table 1* in (*Unal et al., 2015*) | RRID: AB_10000240 |
| Metabotropic glutamate receptor 1a (mGluR1a) | Gp | 1:500-1:1000 | Frontier Institute, mGluR1a-GP-Af660 | Characterized by Tanaka et al. 2000 *Eur. J. Neurosci.* 12, 781–792 | RRID: AB_2531897 |
| Neuronal nitrogen oxide synthase (nNOS) | Rb | 1:1000 | EMD Millipore, AB5380 | Supplementary Table 2 in (*Viney et al., 2013*) | RRID: AB_91824 |
| Parvalbumin (PV) | Rb | 1:1000 | Swant, PV-28 | Supplementary Table 2 in (*Viney et al., 2013*) | RRID: AB_2315235 |
| Parvalbumin (PV) | Gt | 1:1000 | Swant, PVG-214, lot 3.6 | Supplementary Table 2 in (*Viney et al., 2013*) | RRID: AB_2313848 |
| Parvalbumin (PV) | Gp | 1:5000 | Synaptic Systems, 195 004, lot 5 | Supplementary Table 2 in (*Viney et al., 2013*) | RRID: AB_2156476 |
| Purkinje cell protein 4 (PCP4) | Rb | 1:1000 | Santa Cruz, sc-74816, lot G0814 | Characterized by San Antonio et al. 2014 *J. Comp. Neurol.* **522**, 1333–1354 | RRID: AB_2236566 |
| Pro-cholecystokinin (pro-CCK) | Rb | 1:500 | April 2005 gift (similar to Frontier Institute, CCK-pro-Rb-Af350) | Supplementary Table 2 in (*Viney et al., 2013*) | |
| SATB1 (N-14) | Rb | 1:200 | Abcam, ab70004 | Supplementary Table 2 in (*Viney et al., 2013*) | RRID: AB_1270545 |
| SATB1 (N-14) | Gt | 1:200-1:250 | Santa Cruz, sc-5989 | Supplementary Table 2 in (*Viney et al., 2013*) | RRID: AB_2184337 |
| SMI32 (neurofilament H non-phosphorylated) | Ms | 1:1000 | Covance, SMI-32R lot 14835102 | Similar to immunoreactivity characterized in primates by Campbell and Morrison 1989 *J. Comp. Neurol.* 282, 191–205 and in rats by Ouda et al. 2012 *Brain Struc. Func.* **217**, 19–36 | RRID: AB_509997 |
| Substance P receptor (NK1R) | Rb | 1:500 | Millipore, AB5060, lot LV1525037 | Characterized by Shigemoto et al. 1993 *Neurosci. Lett.* **153**, 157–160 | RRID: AB_2200636 |
| Vesicular GABA transporter (VGAT) | Gp | 1:500 | Synaptic Systems, 131 004 | Supplementary Table 2 in (*Viney et al., 2013*) | RRID: AB_887873 |
| Vesicular glutamate transporter 2 (VGLUT2) | Gp | 1:2000 | Synaptic Systems, 135 404 lot 135404/16 | Similar to immunoreactivity characterized in mouse hippocampus by Herzog et al. 2006 *J. Neurochem.* **99**, 1011–1018 | RRID: AB_887884 |
| Wfs1 | Rb | 1:500 | Proteintech, 11558–1-AP | *Table 1* in (*Unal et al., 2015*) | RRID: AB_2216046 |

DOI: https://doi.org/10.7554/eLife.34395.030

(CED). Extracellularly recorded cells in the MSDB were juxtacellularly labeled using 200 ms current pulses (*Pinault, 1996*) followed by a 4–8 hr recovery period (*Table 2*). Unlabeled single neurons were classified as from the MSDB if they were recorded near juxtacellularly labeled cells, or their firing patterns closely matched neurons that were already established as being from the MSDB. A total of 16 animals were used for analysis (*Table 7*). Two other animals containing MS neurons with axon projecting to the EC were excluded because the axons could not be matched to a recorded neuron. Selection of MSDB neurons for recording was initially random. After recovery of the first three neurons with EC-projecting axons (TV50a, TV58g, MS14i), subsequent experiments were biased towards targeting neurons in the MS that had similar rhythmic bursting firing patterns to these three neurons; most other neurons were bypassed. Therefore, other kinds of EC-projecting GABAergic neurons may not have been recorded. Animals with recorded neurons projecting to other parts of the temporal cortex and with firing patterns that differ from orchid cells will be reported in detail in planned future studies. The CA3-projecting Teevra neurons (*Figure 5f*) are taken from a separate study (*Joshi et al., 2017*). In experiments with juxtacellular labeling in the EC (n = 8 mice; animals TV92, TV108, TV111-116), the recovery period was ~1 hr.

## Tissue processing

Mice were deeply anesthetized with sodium pentobarbital (50 mg/kg, i.p.) and transcardially perfused with 0.1 M phosphate buffer (PB) followed by 4% paraformaldehyde 15% v/v saturated picric acid, 0.05% glutaraldehyde in 0.1 M PB at pH 7.4. Some brains were postfixed overnight in glutaraldehyde-free fixative. After washing in 0.1 M PB, 70–100 μm coronal sections were cut using a Leica VT 1000S vibratome (Leica Microsystems) and stored in 0.1 M PB with 0.05% sodium azide at 4°C. Streptavidin-conjugated fluorophores were used to visualize neurobiotin-labeled neuronal processes within tissue sections previously permeabilized by Tris-buffered saline (TBS) with 0.3% Triton X-100 (TBS-Tx) or through rapid 2x freeze-thaw (FT) over liquid nitrogen (cryoprotected in 20% sucrose in 0.1 M PB). For light microscopic visualization, analysis, and 3D neuronal reconstruction, TBS-Tx- or FT-processed sections were processed using horseradish peroxidase (HRP)-based diaminobenzidine (DAB) reactions as previously described (*Viney et al., 2013*).

## Immunohistochemistry

For the molecular identification of labeled neurons and their postsynaptic target neurons, immunohistochemistry was carried out as previously described (*Unal et al., 2015*; *Viney et al., 2013*). Specificity information for primary antibodies is in *Table 11*. To test the immunoreactivity of multiple markers on the same neurons (e.g. on postsynaptic targets of septo-cortical neurons), we employed an iterative strategy based on area-dependent marker frequency, subcellular localization, colocalization probability, antibody species, and fluorophore. Typically, we tested rare markers first, along with non-cytoplasmic markers (e.g. nucleus or cell membrane) to maximize the available subcellular domains for testing. Multiple fluorescence channel immunoreactivity was documented with epifluorescence or confocal microscopy (see below) followed by demounting the sections, washing in TBS or TBS-Tx, and repeating the procedure with different antibodies. Cytoplasmic markers, such as calcium binding proteins, were often tested last. In each round of immunohistochemistry, 'negative' controls were included that lacked the primary antibodies, along with 'positive' controls from a different brain that included the primary antibodies. In the next round of immunohistochemistry, the previous positive controls became new 'negative' controls by lacking the new primary antibodies. Thus, the number of control sections increased with each round of processing, and each control was compared to the test section, along with a comparison of images acquired in the same location before and after each round of immunohistochemistry. Undetectable immunoreactivity within the neuron for a given fluorescence channel was excluded for that particular round if no new signals were present in the vicinity of the neuron of interest. A rare marker recognized by a particular primary antibody was tested first if the antibody was raised in the same species as a primary antibody for a common marker to avoid saturation by binding to the existing secondary antibody.

## Microscopy

Confocal microscopy (Zeiss LSM 710 with ZEN software) was used to document identified neurons and their targets, as previously described (*Unal et al., 2015*; *Viney et al., 2013*). Overviews of multi-

channel multi-round sections tested with immunohistochemistry were acquired with widefield epifluorescence either on the same microscope as used for confocal imaging (with Axiovision software), or on a Leitz DMRB microscope (Leica). Electron microscopy was carried out as previously described (*Unal et al., 2015*; *Viney et al., 2013*).

## Electrophysiological data analysis

Data were analyzed in Mathematica (Wolfram Research), MATLAB (MathWorks) and Spike2 (CED). Movement periods were detected by the combination of video, wheel activity and in some cases EMG or accelerometer. Whisking, postural shifts and respiratory rate were qualitatively observed from the video. For labeled neurons, only data acquired before juxtacellular labeling were used for analysis. Teevra cell firing patterns were taken from 13 identified (recorded then juxtacellularly labeled) CA3-projecting cells reported in *Table 1* of (*Joshi et al., 2017*), with their 'RUN' defined here as locomotion (LM) and 'REST' defined as immobility (IM). Note that most Teevra cells were recorded from fully trained head-restrained mice under goal-directed movement conditions (e.g. running for a sucrose reward).

### Local field potentials

The position of the CA1d LFP recording was estimated based on the polarity of sharp waves (*Buzsáki, 1986*) and the presence of ripples (*Buzsáki et al., 2003*). Both strata oriens and pyramidale contained positive sharp waves, and stratum radiatum contained negative sharp waves. The upper part of superficial stratum pyramidale consisted of both positive and negative sharp waves. In stratum pyramidale, pyramidal cells were often recorded. In one case (animal TV58), a hippocampal neuron weakly labeled at the recording site was used as confirmation. In addition to polarity of the SWRs, in three cases (AJ44j, AJ50h, AJ50j), mid gamma (55–80 Hz; detection threshold:>1 s.d. above the mean cycle amplitude) coupling to the peak of theta oscillatory activity was also used to estimate the location of LFP recording in stratum pyramidale. Mid-gamma (55–80 Hz) is coupled to the peak of CA1 pyramidale theta oscillatory activity (*Colgin et al., 2009*; *Lasztóczi and Klausberger, 2014*; *Schomburg et al., 2014*).

### Theta and gamma

Theta periods were detected by filtering the CA1d LFP for theta (5–12 Hz) and delta (2–4 Hz) and computing a power ratio. For cell MS74b, theta (and gamma, below) was detected from an electroencephalogram (EEG) at 2.10 mm posterior and −2.50 mm lateral of Bregma, which was coherent with CA1d strata oriens/pyramidale. Theta phase was calculated by linear interpolation between troughs of the band-pass filtered theta oscillations, with 0° and 360° set as the troughs. The Rayleigh test was used to test for uniformity of circular phase distributions. Mean phase and mean vector length were used as measures of the preferred phase and coupling strength, respectively, both for the spike-theta and gamma troughs-theta coupling. Mid-gamma (55–80 Hz) troughs were detected from the CA1d LFP for the entire recording period (detection threshold:>1 s.d. above the mean cycle amplitude), with most gamma troughs detected during movement-related theta oscillations. If cross-frequency coupling was observed (gamma troughs coupled to theta peak), spike coupling was measured for gamma cycles as described for theta above. The troughs of EC theta cycles (detected from the 5–12 Hz filtered EC LFP) were also tested for their coupling to CA1d theta oscillations, as described above. Phase histograms were smoothed by convolving with a Gaussian.

### Large amplitude irregular activity

To detect zero-crossings of large amplitude irregular activity (LIA) in the CA1d LFP (oscillations < 5 Hz; 'non-theta'), the wideband LFP was low-pass filtered in Spike2 by smoothing (0.08 s window), followed by DC removal (0.2 s window). The falling level zero-crossings of the cycles were detected with a minimum interval of 0.2 s.

### Sharp-wave associated ripple oscillations

The power of the 130–230 Hz band-pass filtered CA1d LFP was used to detect SWRs, with a threshold of at least 4 s.d. above the mean power. Firing rate changes during SWRs were compared to 1000 shuffled firing rate distributions of periods outside SWRs (excluding periods with detected

theta oscillations), as previously described (*Katona et al., 2014*) (their Method 1). Neurons with less than 20 detected SWRs were excluded.

## Firing patterns

Mean firing rates were calculated in 1 s windows within each behavioral state. The LM-IM index (*Figure 5*) was calculated by normalizing mean firing rates [(LM - IM)/(LM + IM)], with a positive value indicating an increase in mean firing rate during LM and a negative value indicating a decrease in mean firing rate during LM. Bursts were defined as a train of >3 spikes with interspike intervals (ISIs) < 40 ms. Burst incidence was defined by the total number of bursts in 1 s windows. Interburst intervals were calculated by measuring the time elapsed between the first spikes of consecutive bursts.

Time-frequency plots (*Figure 5—figure supplement 1*) used the ContinuousWaveletTransform in Mathematica (Morlet wavelet, 1 kHz sampling rate, 12 octaves, 16 voices).

## Speed measurements

The circular treadmill (*Table 7*) consisted of a polyurethane foam cylinder of 146 mm diameter and 125 mm length, rotating on an axle supported on one side. On the axle was a disc for an optical quadrature encoder, providing 2000 edges per revolution (sum of rising and fall edges of channels A and B). The optical encoder was connected to an Arduino Uno microcontroller running our own code, and outputting through a 12-bit 2-channel digital to an analogue converter (DAC). The Arduino program generated a counter for the wheel movement, the counter increasing every time the wheel made 1/2000th of a rotation in the forward-running direction, and decreasing for the backward direction. The output consisted of two signals, speed and movement period, 0 to +4 V, that were re-digitized by the Power1401 and recorded concurrently with electrophysiological signals in Spike2. The speed signal was calculated at 10 ms intervals, using the distance traveled by the wheel as recorded by the counter, divided by time elapsed since the last calculation. The values sent to the DAC were calibrated so that 1 V output signal represented a speed of 0.6912 m/s at the cylinder surface. The movement period signal was raised to 4 V when the counter increased by two compared to a value 10 ms earlier, and dropped to 0 V if the counter increased by less than 11 compared to a value taken 50 ms earlier. Speed was not measured for all mice.

## Neuronal reconstructions and delineation of cortical areas

Allen Brain Atlas images of selected marker genes expressed in coronal mouse brain sections were used as reference sections to define the positions of labeled orchid cell axons within processed fluorescence and DAB-reacted sections (*Table 5*, *Figure 3—figure supplement 2*, *Figure 3—figure supplement 3*, *Figure 3—figure supplement 4*). Reconstruction of neuronal processes within each series of DAB-reacted sections were carried out using Neurolucida (MBF Bioscience) in 3D, as previously described (*Viney et al., 2013*). For the partial cortical axon reconstruction of TV58g (*Figure 3*, *Figure 3–video 1*), regional, sub-regional and laminar boundaries were assigned based on key Allen Brain Atlas gene expression profiles (*Table 5*) and a series of DAB-reacted sections immunoreacted for calbindin (*Figure 3—figure supplement 3d*). The reconstructed axon was scaled up to the original unprocessed (freshly sectioned) z-thickness. Reconstructions of somata, dendrites, and local axon in the medial septum were carried out on a drawing tube in 2D (Leitz Dialux 22 microscope). Reconstructions are available at http://neuromorpho.org/.

## Quantification of virally-labeled axons

The axonal collateralization of MSDB neurons projecting to the EC was compared between extrahippocampal areas and the hippocampus. Sections containing the PrSd, RSg, CA1, dentate gyrus and SUB that could be delineated within the same section were sampled from animals with anterogradely labeled EYFP+ MSDB axons (n = 3 animals, three sections each; 70–100 μm thickness per section). Representative regions of interest were defined (ROIs; width 19.5 μm, height spanning all the layers in each cortical area; n = 3 samples per region per section) and all EYFP+ axons that crossed both lateral edges of the ROI were counted using ImageJ (n = 120 ROIs; n = 274 axons). The proportion of axons per brain region of the total in each section was calculated and compared using the Kruskal-Wallis test.

Virally-labeled EYFP+ axon terminals from three mice (MS60, MS66, MS76) were tested for their immunoreactivity to VGAT in the PrSd and RSg. A sample of 2–3 70 µm-thick sections from each brain were used for quantification. For each section, an LSM710 confocal microscope (with Axio Imager.Z1, Carl Zeiss) with ZEN 2008 software v5.0 (Zeiss) was used to acquire two-channel fluorescence z-stacks at 40x magnification. Using the ImageJ Cell Counter plugin, all EYFP+ axon terminals within the stacks were counted, and classified as VGAT immunopositive or lacking detectable VGAT immunoreactivity. The latter classification was assigned if other VGAT-immunopositive puncta were imaged nearby within the same optical section (100% of cases).

## Stereological counting

Three adult male C57BL6/J mice (28–30 g) were anesthetized with a terminal dose of pentobarbitone and perfuse-fixed (4% PFA and 0.05% glutaraldehyde) followed by post-fixation overnight (0.4% PFA). Brains were sectioned coronally at 50 µm and collected in series from a randomly selected point before the MSDB structure.

A random starting section was chosen for each brain, and every third section thereafter was selected, to cover the entire MSDB, resulting in 8–9 sections per animal. Sections were permeabilized in TBS-Tx and blocked for 1 hr in 20% normal horse serum (NHS) in TBS-Tx at room temperature, then incubated in a primary antibody solution containing TBS-Tx and 1% NHS for 3 nights at 2–8°C. Sections were washed then incubated in a secondary antibody solution containing TBS-Tx and 1% NHS for 4 hr at room temperature. They were then washed, incubated in a DAPI solution (1:1000 in 0.1M PB) for 60 s, washed and mounted on slides in Vectashield. Antibodies used (*Table 11*): guinea pig anti-mGluR1a, goat anti-SATB1, rabbit anti-PV, donkey anti-guinea pig Alexa Fluor 488 1:1000, donkey anti-goat Cy3 1:400, donkey anti-rabbit Alexa Fluor 647 1:250.

The optical fractionator method was used to carry out systematic random sampling. Stereo Investigator (MBF Bioscience) was used for obtaining images and stereological counting. Each section was scanned using a fully motorized Axio Imager M.2 fluorescence microscope, with a 63 × 1.4 NA oil objective. The probe depth was 5 µm, with a 2 µm guard zone above, and an interval of 1 µm. The grid size was 240 × 160 µm, and the counting frame was 120 × 80 µm. Segment contours were drawn using a 10x or 5x objective, using the SATB1 marker for a guide to delineate the MSDB region. The mounted section thickness was set to 50 µm for all sections.

For counting, markers were used for the seven categories of PV only, mGluR1a only, SATB1 only, PV and SATB1, PV and mGluR1a, mGluR1a and SATB1, and PV, mGluR1a and SATB1. The point at which the nucleolus first appeared from above was the point at which the neuron was counted. DAPI was used to identify this for mGluR1a-positive cells. Only neurons immunopositive for at least one marker were counted. Nucleoli which touched the green inclusion line of the counting frame were counted, but not counted if they touched the red exclusion line. In cases where the contour intersected a neuron, it was counted if more than half of the nucleolus fell within the contour boundary.

For statistics, we used the Gundersen coefficient of error (m = 1). For all markers within each brain, the errors were 0.05, 0.06 and 0.05 (n = 3 mice). Potential sources of error include antibody penetration for PV and mGluR1a, such that counts of these two markers may be slightly underrepresented.

## Statistics and sample size

All statistical tests were run in Mathematica or MATLAB and are stated in the text. In all cases we used an $\alpha$ level of 0.05. The permutation test used 10,000 permutations. The circular statistics toolbox in MATLAB was also used (P. Berens, CircStat: A Matlab Toolbox for Circular Statistics, Journal of Statistical Software, Volume 31, Issue 10, 2009, http://www.jstatsoft.org/v31/i10). Sample sizes for each experiment are stated in the text. Initially, neurons were recorded randomly by slowly advancing the glass electrode through the medial septum, and a subset were juxtacellularly labeled in order to determine their cortical target region(s) *post hoc*. In later experiments, neurons with specific firing pattern signatures (see Results, 'Behavioral and network state-dependent firing of orchid cells') were targeted while all other neurons were ignored. We aimed to have a standard deviation of <10 Hz for the mean firing rates during immobility for orchid cells. Neurons with firing patterns and projections that differed from orchid cells and Teevra cells will be reported in detail elsewhere (including the neurons represented in *Table 10*).

## Additional details of orchid cell projections

The first collateral of orchid cell TV58g specifically innervated a sub-region of FC medial to SUBd and ventral to the caudal end of the corpus callosum (*Figure 3—figure supplement 4a,b*). A faint thin axon, originating from the MSDB near the labeling site of cell TV50a projected to the medial CA2/FC region (identified by *Amigo2* gene expression, *Figure 3—figure supplement 4*) but could not be proven to originate from the main projection axon. Cells MS19b and AJ50j projected along the dorsal fornix near the midline similar to TV58g (*Figure 3* ), but TV50a, MS14i, AJ50h, AJ44j and MS74b orchid cell axons traveled along the lateral dorsal fornix, passing CA1d, before heading to SUBd/PrSd. A single branch from the main axon of MS14i in the fornix crossed into CA1 stratum lacunosum, but no varicosities were observed until the branch traveled temporally and innervated the SUBd. AJ50j innervated two separate regions of RSg, one was more septal (layer 6), other was temporal (deep and superficial layers) (*Figure 3—figure supplement 2*, *Figure 3—figure supplement 3*). The axons of AJ44j and MS74b were weakly labeled: the main axons faded away within the angular bundle adjacent to the PrSd.

## Acknowledgements

We thank Linda Katona for help with physiological analysis; Kristina Wagner for help with tissue processing and electron microscopy; Michael Howarth, Katja Hartwich, Amar Sharma, Kathryn Newton, Szabolcs Biro, Eszter Kormann and Laszlo Marton for technical assistance; Ben Micklem for advice on reconstructions, stereology, and for building the speed monitor; Thomas Forro for advice on recording in head-restrained mice. We thank Balint Lasztóczi and A Tugrul Ozdemir for comments on an earlier version of the manuscript. We thank Lynn Enquist at Princeton University for stocks of PRV-hSyn-Cre, and David Dupret and Pavel Perestenko for aliquots of AAV. We also thank M Watanabe for the pro-CCK antibody.

## Additional information

### Funding

| Funder | Grant reference number | Author |
|---|---|---|
| Medical Research Council | MC_UU_12024/4 | Tim James Viney<br>Minas Salib<br>Abhilasha Joshi<br>Gunes Unal<br>Naomi Berry<br>Peter Somogyi |
| Wellcome | 108726/Z/15/Z | Tim James Viney<br>Minas Salib<br>Abhilasha Joshi<br>Gunes Unal<br>Naomi Berry<br>Peter Somogyi |
| Felix Doctoral Scholarship | | Abhilasha Joshi |

The funders had no role in study design, data collection and interpretation, or the decision to submit the work for publication.

### Author contributions

Tim James Viney, Conceptualization, Data curation, Formal analysis, Supervision, Validation, Investigation, Visualization, Methodology, Writing—original draft, Writing—review and editing; Minas Salib, Data curation, Formal analysis, Investigation, Visualization, Methodology, Writing—review and editing; Abhilasha Joshi, Data curation, Formal analysis, Validation, Investigation, Visualization, Methodology, Writing—review and editing; Gunes Unal, Data curation, Formal analysis, Investigation, Visualization, Writing—review and editing; Naomi Berry, Data curation, Investigation, Visualization, Methodology, Writing—review and editing; Peter Somogyi, Conceptualization, Resources,

Data curation, Formal analysis, Supervision, Funding acquisition, Validation, Investigation, Visualization, Methodology, Writing—review and editing

### Author ORCIDs
Tim James Viney http://orcid.org/0000-0001-6444-1188
Minas Salib http://orcid.org/0000-0001-9938-7978
Abhilasha Joshi http://orcid.org/0000-0002-0511-3747
Gunes Unal http://orcid.org/0000-0003-3013-0271

### Ethics

Animal experimentation: All procedures involving experimental animals were approved by the Department of Pharmacology Animal Welfare and Ethical Review Body under approved personal and project licenses (project licence number: 30/3240) in accordance with the Animals (Scientific Procedures) Act, 1986 (UK) and associated regulations. All surgery was performed under isoflurane anesthesia with a peri-operative dose of buprenorphine, and every effort was made to minimize suffering.

### Decision letter and Author response

Decision letter https://doi.org/10.7554/eLife.34395.033
Author response https://doi.org/10.7554/eLife.34395.034

## Additional files

### Supplementary files

• Transparent reporting form
DOI: https://doi.org/10.7554/eLife.34395.031

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
