## [Decision Letter]

Thank you for sending your article entitled "Shared rhythmic subcortical GABAergic input to the entorhinal cortex and presubiculum" for peer review at *eLife*. Your article is being evaluated by three peer reviewers, one of whom is a member of our Board of Reviewing Editors and the evaluation is being overseen by Gary Westbrook as the Senior Editor.

Given the rather long list of major essential revisions, the editors and reviewers invite you to respond within the next two weeks with an action plan and timetable for the completion of the additional work. We plan to share your responses with the reviewers and then issue a binding recommendation.

In particular, it is most important that you provide clarification about the exact quantitative criteria that justified classification of the "Orchid" cell type. The reviewers agreed that it is not possible to fully evaluate the significance of these findings until the data are presented more transparently and analyzed more thoroughly. Please find the specific concerns and requests of the reviewers outlined below.

Essential revisions:

1) A large part of the evidence for specificity of the projection to the EC comes from the experiment described in Figure 1. However, this experiment lacks adequate quantification. In particular, to support the claim that only sparse axons collaterals target the hippocampus, while labeling is found in other extra-hippocampal areas, a more rigorous analysis is required. Similarly, the statement that "A subset were immunoreactive for vesicular GABA transporter (VGAT, Figure 1C)" is anecdotal and requires proper quantification. For this experiment, it's also important to show where in the EC the axons terminate. For example, Fuchs et al. suggest that septal PV axons preferentially target layer II, whereas Gonzalez-Sulser et al., indicate axons are enriched in layers II-V.

2) It is unclear how many neurons (and animals) were recorded in total in order to identify the 8 labeled neurons reported. This should be clearly stated. It's also unclear how the 8 labeled neurons were selected. This should also be clearly stated. Were these the only neurons with axons projecting towards the EC, or were additional selection criteria applied? If this population matches the "21% of the virally-labeled MS neurons" (subsection “GABAergic orchid cells project to multiple extra-hippocampal regions”), then it should also make up about 20% of the recorded neurons that project to the EC, and a much smaller percentage of the total number of recorded neurons. If the percentage is much greater, or less, then what is the reason for this? This could substantially affect how one interprets the data.

3) The authors need to better explain to readers how the neurons that they have identified relate to previously identified populations of MS neurons. For example, in the passage in subsection “Medial septal neurons projecting to the entorhinal cortex”, the authors should explain why they tested for PV, mGlur1a, and SATB1 (e.g., what does SATB1 immunoreactivity indicate in the medial septum with regard to cell classification?). Also, the authors should clarify which cell groups correspond to cells that have been classified previously (as they do for "Teevra cells") and which correspond to cells of unknown classification. Similarly, in subsection “Preferential synaptic targets of orchid cells”, the authors write, "Targets in three regions…. Were SATB1+… but CCK-…" They should explain what this means, if anything. Does this point toward a particular type of cell? Another example of this is in subsection “MSDB neurons immunopositive for PV and/or SATB1 and/or mGluR1a”, where the authors write, "The largest group comprised PV-/SATB1+/mGluRa- neurons". What type of cells are these? Do they correspond to cells that have been reported previously?

4) With regard to Figure 2, the authors state that 5/8 neurons formed thin collaterals/terminals in deep layers of caudo-dorsal EC (subsection “Firing patterns of septo-entorhinal neurons”). What about the other 3 neurons?

5) The description of the dendrites of labeled MS neurons is anecdotal (subsection “GABAergic orchid cells project to multiple extra-hippocampal regions”). It would be strengthened by quantitative analysis and comparison to other populations of MS neurons.

6) The authors write, "We determined the molecular profiles of 13 targeted cortical neurons and used electron microscopy to test the reliability of predicting synaptic connections from light microscopy" (subsection “Preferential synaptic targets of orchid cells”). Please state from how many mice. This already seems a small number of representative neurons from which to draw a strong conclusion, and maybe even less so if from few mice.

7) The authors write, "The axon showed target selectively, as nearby interneurons with different molecular profiles were not targeted" (subsection “Preferential synaptic targets of orchid cells”). Again, proper quantification is needed here.

8) The introduction of 8 additional neurons in subsection “Behavioral and network state-dependent firing of orchid cells” raises additional questions that require clarification. What are the false positive and false negative rates for the preceding criteria? In other words, when these criteria are applied to all neurons with recovered axons (including those that have not been reported), how many meet the criteria and don't have EC projecting axons, and how many don't meet the criteria but do have EC projecting axons / other properties of 'orchid cells'?

9) In subsection “Behavioral and network state-dependent firing of orchid cells”, it is unclear what the authors meant in the following passage: "The first peak was at 143.8[…] during locomotion[…] but occurred later during the other states". What exactly does "the first peak" refer to?

10) In subsection “Behavioral and network state-dependent firing of orchid cells”, what is meant by "due to state-dependent features of the spike bursts"?

11) The results described in subsection “Behavioral and network state-dependent firing of orchid cells” are not entirely clear. At one point, the authors refer to "the identified CA3-projecting Teevra cells". Which ones? Do the authors mean the 2 cells that are mentioned in this subsection?

12) In the Materials and methods section, the authors write, "γ power was also used to estimate the location of the LFP recording". How exactly?

13) In the Materials and methods section, the authors write, "If cross-frequency coupling was observed". How was this defined?

14) In the Materials and methods section, the authors write, "Inter-burst intervals were calculated by taking the timing of the first spike of each burst". This method explanation does not make sense.

15) The experiment described in Figure 6 (subsection “Relationship between entorhinal and hippocampal oscillations”) is quite confusing. It states that 1 pyramidal cell was recorded in EC to confirm the recording location – does this imply that the data reported is only from 1 mouse? When talking about MS firing rate distributions being non-Poisson, is this referring to 9/9 neurons from the same mouse? More generally, the point of these experiments is unclear. How do they go beyond similar experiments from the Buzsaki lab (e.g. Chrobak and Buzsaki, 1998; Mizuseki et al., 2009)?

16) The data in Figure 7 would be more useful if they could be more directly compared to the cell labeling data. For example, is the proportion of triple positive neurons similar to the proportion of 'orchid cells' in the total population of recorded MS neurons? And, can the data in Figure 1fFbe normalized to the total cell populations in Figure 7?

17) Overall the organization of the manuscript makes it harder to follow than it perhaps needs to be. For example, would it be clearer if when describing the labeled neurons if their anatomy and molecular markers were described first, and then the electrophysiology was built upon from this? At present, the manuscript jumps back and forth between anatomy and electrophysiology.

[Editors' note: further revisions were requested prior to acceptance, as described below.]

Thank you for resubmitting your work entitled "Shared rhythmic subcortical GABAergic input to the entorhinal cortex and presubiculum" for further consideration at *eLife*. Your revised article has been evaluated by Gary Westbrook (Senior editor) and a Reviewing editor.

The manuscript has been improved, but there are some remaining issues that need to be addressed as outlined below. The primary concern with the previous submission, namely that insufficient details and quantification were provided regarding the characterized orchid cells, has been satisfactorily addressed. However, not surprisingly given that the paper was so substantially revised, there are some minor issues that remain to be remedied. When this manuscript is resubmitted, you should either consistently mark all changes that were made or submit a clean version of the manuscript.

1) In the Introduction, "such as a place cell sequences" should be "such as place cell sequences".

2) In subsection “Medial septal neurons projecting to the entorhinal cortex”, add "for PV, SATB1, and mGluR1" after "triple immunopositive" to avoid confusion.

3) Insubsection “GABAergic orchid cells project to multiple extra-hippocampal regions”, the authors write, "mid-γ oscillations were coupled to theta cycles preferentially at the theta peak (n = 7/8 neurons)". This is confusing because γ was not recorded intracellularly in this study. So, clarification is needed to explain why measurements are given with regard to numbers of neurons.

4) In subsection “Behavioral and network state-dependent firing of orchid cells”, "different in rates" should be "different than rates".

5) In subsection “MSDB neurons immunopositive for PV and/or SATB1 and/or mGluR1a”, the authors use the word "inhibited", but it would perhaps be better to use a different word choice because "inhibited" implies that IPSPs or IPSCs were actually measured.

6) In the Discussion section, the authors write, "orchid cells represent a distinct subpopulation of Komal cells", but this seems incorrect because the authors stated that orchid cells fire at the theta peak and Komal cells fire at the theta trough.

7) In the Discussion section, the authors should add, "in the present study" at the end of the sentence for the purpose of clarity.

8) In the Materials and methods section, the authors should replace "elsewhere" with "in planned future studies".

9) In the Materials and methods section”, "remove" should be "removal".

10) In the Materials and methods section, providing more detail (i.e., the width of the wavelet) would be helpful.

11) In Table 2, it is unclear what the letters a-k mean. This should be explained in the caption.

---

## [Author Response]

Thank you for sending your article entitled "Shared rhythmic subcortical GABAergic input to the entorhinal cortex and presubiculum" for peer review at eLife. Your article is being evaluated by three peer reviewers, one of whom is a member of our Board of Reviewing Editors and the evaluation is being overseen by Gary Westbrook as the Senior Editor.Given the rather long list of major essential revisions, the editors and reviewers invite you to respond within the next two weeks with an action plan and timetable for the completion of the additional work. We plan to share your responses with the reviewers and then issue a binding recommendation.In particular, it is most important that you provide clarification about the exact quantitative criteria that justified classification of the "Orchid" cell type. The reviewers agreed that it is not possible to fully evaluate the significance of these findings until the data are presented more transparently and analyzed more thoroughly. Please find the specific concerns and requests of the reviewers outlined below.

To describe quantitative criteria for classifying these rhythmic medial septal neurons as ‘orchid cells’, we present our answers to Questions 2, 3 and 8. To summarise these points, from 113 implanted mice (with typically 2-3 recording days per mouse comprising hundreds of single-neuron extracellular recordings during movement and rest behaviour) we recovered 10 medial septal neurons with EC-projecting axons, with 8 defined here as ‘orchid cells’. We have found other kinds of medial septal projection neurons, but the strongly labelled ones in our database are too few in number to recognise patterns and it will still take months to years to analyse them.

The 8 labelled EC-projecting neurons in this manuscript probably represent the most comprehensively defined group of subcortical neurons reported to date, since they not only have firing patterns that we recognised as different from another group of medial septal cells we reported recently (CA3-projecting Teevra cells; Joshi et al., 2017), but they also project to specific extra-hippocampal regions (mainly the dorsal presubiculum *and* caudo-dorsal entorhinal cortex).

These extra-hippocampal regions have been carefully studied in terms of their local circuits and corticocortical projections in behaving rodents (e.g. interneurons and principal cells, Tukker et al., 2015; Preston-Ferrer et al., 2016; Burgalossi et al., 2014; Schmidt-Hieber and Häusser, 2013; Boccara et al., 2010), neuroanatomy/electrophysiology (e.g. Ray et al., 2017; Insausti et al., 1997; Nassar et al., 2015) and in terms of their modulation by pathway-selective inputs (e.g. thalamocortical, Simonnet et al., 2017; medial septal GABAergic inputs, Fuchs et al., 2016; Gonzalez-Sulser et al., 2014;medial septal glutamatergic inputs, Justus et al., 2017). Our discovery and combined neuroanatomical/physiological definition of orchid cells advances the field by providing data on a specific group of subcortical neurons and circuits involved in cortical neuronal coordination, particularly theta and mid-γ frequency rhythmic activity, required for spatial navigation and mnemonic processes that involve extra-hippocampal circuits (e.g. Koenig et al., 2011; Jeffery et al., 1995; Brandon et al., 2013).

Essential revisions:1) A large part of the evidence for specificity of the projection to the EC comes from the experiment described in Figure 1. However, this experiment lacks adequate quantification. In particular, to support the claim that only sparse axons collaterals target the hippocampus, while labeling is found in other extra-hippocampal areas, a more rigorous analysis is required. Similarly, the statement that "A subset were immunoreactive for vesicular GABA transporter (VGAT, Figure 1C)" is anecdotal and requires proper quantification. For this experiment, it's also important to show where in the EC the axons terminate. For example, Fuchs et al. suggest that septal PV axons preferentially target layer II, whereas Gonzalez-Sulser et al. indicate axons are enriched in layers II-V.

We have quantified axon collaterals in the hippocampus/dentate gyrus and extra-hippocampal areas, which is now reported in the Results section and Materials and methods section:

“We quantified the distribution of axonal collaterals in hippocampal and extra-hippocampal cortical regions (n=3 mice; 3 sections per animal). The proportion of axonal branches in the dorsal presubiculum (PrSd) and granular retrosplenial cortex (RSg) (median: 37%; interquartile range (IQR): 34.3% – 47.2%) was substantially greater than in CA1, the dentate gyrus and the dorsal subiculum (SUBd) (median: 5%; IQR: 2.4% – 9%), where only rare axonal collaterals were observed (*P*=1.6 x 10^-7^, Kruskal-Wallis test)”.

We have also quantified immunoreactivity for VGAT in virally-labelled axons (details added to Materials and methods section) and now state in the Results section:

“Within the dorsal presubiculum (PrSd) and granular retrosplenial cortex (RSg), 71.0 ± 25.9% of EYFP+ axonal terminals (mean ± s.d, n = 1046/1416 counted terminals within 12 sampled areas from 3 mice) were immunoreactive for vesicular GABA transporter (VGAT, Figure 1C).”

We have now revised Figure 1—figure supplement 1 to show in both coronal and horizontal sections where the virally-labelled axons terminate in the EC (panels a and b). Gonzalez-Sulser et al., 2014 reported that ~42% of virally-labelled neurons from the mouse MS/(DB) projecting to the *medial* EC were GABAergic. We found a high proportion of VGAT-immunoreactive terminals in the PrSd and RSg from collaterals of axons projecting to the *caudo-dorsal* EC. Gonzalez-Sulser et al., found that PV was in a minority of the mEC-projecting neurons (<10%). Thus, the enrichment they observed in layers II and V of mEC may represent mostly non-PV medial septal axons. Fuchs et al., 2016 showed that the PV+ subset of GABAergic MSDB axons target primarily layer II of mEC. The results on orchid cells complement these medial EC-related studies with a different approach, since we used retrograde labelling from the EC (PRV-hSyn-Cre) and anterograde Cre-dependent labelling in the MSDB (EYFP+) to restrict the population of labelled neurons to the injected caudo-dorsal EC and to find their collaterals en route to the EC. We now show in Figure 1—figure supplement 1 that axons projecting to the caudal EC terminate in multiple layers, including deep layers. PV+ neurons were 24.2% of the virally-labelled MSDB neurons (Figure 1F).

We have added the following to the Results section: “Virally labeled medial septal axons with extensive terminals were observed in all layers of the EC (n=4 mice, from 18 coronal or horizontal sections), with collaterals traveling both radially and horizontally (Figure 1—figure supplement 1A,B)” and have revised Figure 1—figure supplement 1 to show greater detail of medial septal terminals in the EC.

We also add the following to the Discussion section: “GABAergic MSDB neurons projecting to the medial EC preferentially innervate layers II and V (Gonzalez-Sulser et al., 2014a), and the small PV+ subpopulation of GABAergic MSDB neurons provides a preferential input to ‘fast spiking’ interneurons in layer II (Fuchs et al., 2016).”

2) It is unclear how many neurons (and animals) were recorded in total in order to identify the 8 labeled neurons reported. This should be clearly stated. It's also unclear how the 8 labeled neurons were selected. This should also be clearly stated. Were these the only neurons with axons projecting towards the EC, or were additional selection criteria applied? If this population matches the "21% of the virally-labeled MS neurons" (subsection “GABAergic orchid cells project to multiple extra-hippocampal regions”), then it should also make up about 20% of the recorded neurons that project to the EC, and a much smaller percentage of the total number of recorded neurons. If the percentage is much greater, or less, then what is the reason for this? This could substantially affect how one interprets the data.

113 mice were implanted with head plates and, in each recording session, several MS neurons were extracellularly recorded with glass electrodes. This was immediately followed by the challenging juxtacellular labelling attempt while the mouse was moving or resting. The above number of mice excludes animals with labelled CA3-projecting ‘Teevra’ neurons (Joshi et al., 2017). For the revised manuscript we recorded from an additional 7 mice in order to target the EC, giving a total of 120 head-plate implanted mice.

The first experiments with recovery of axon in EC (TV50a, TV58g, MS14i) enabled us to search for neurons with similar firing pattern signatures in all subsequent experiments (e.g. burst incidence and preferred theta phase). We therefore selectively targeted the kinds of neurons that were selected for juxtacellular labelling. Our criteria for selection were thus the firing patterns of the first 3 EC-projecting neurons (Table 1).

We have now added the following to subsection “in vivo extracellular recordings and juxtacellular labeling”: “Selection of MSDB neurons for recording was initially random. After recovery of the first 3 neurons with EC-projecting axons (TV50a, TV58g, MS14i), subsequent experiments were biased towards targeting neurons in the MS that had similar rhythmic bursting firing patterns to these 3 neurons; most other neurons were bypassed. Therefore, other kinds of EC-projecting GABAergic neurons may not have been recorded.”

EC-projecting neurons were difficult to target. From the 113 mice, we found only 2 other labelled EC-projecting neurons in addition to the 8 reported in the manuscript. These 2 neurons were excluded because the axon could not be matched to a recording from the multiple attempts in the same animal. Neuron ‘TV46i’ had similar firing patterns to the 8 reported cells (based only on a short recording time), and a projection axon in this animal faded close the EC; 1 collateral was observed in the presubiculum. However, other MS cells were labelled in this animal, so the recording could not unequivocally be matched to the axon. An axon in animal ‘MS98’ projected via a different route (ventral fimbria) to the EC but due to other recorded and labelled neurons in this animal, the axon could not be matched to one recording. Neurons projecting elsewhere in the cortex require further analysis which is very demanding. For most of the unreported labelled neurons, it was not possible to follow the axons to their target regions due to weak labelling. We have updated subsection “in vivo extracellular recordings and juxtacellular labeling” accordingly.

We have removed the sentence about the ‘matching 21% of virally-labelled MS neurons’ from the Results section and revised the Discussion section accordingly: “We found the population of PV+/SATB1+/mGluR1a+ neurons within the MSDB to be relatively small, but the viral tracing revealed that a proportion of these innervated the EC, including those within the rostro-dorsal MS. It remains to be determined what other kinds of MSDB neurons apart from GABAergic orchid cells express these 3 markers”.

3) The authors need to better explain to readers how the neurons that they have identified relate to previously identified populations of MS neurons. For example, in the passage in subsection “Medial septal neurons projecting to the entorhinal cortex”, the authors should explain why they tested for PV, mGlur1a, and SATB1 (e.g., what does SATB1 immunoreactivity indicate in the medial septum with regard to cell classification?). Also, the authors should clarify which cell groups correspond to cells that have been classified previously (as they do for "Teevra cells") and which correspond to cells of unknown classification. Similarly, in subsection “Preferential synaptic targets of orchid cells”, the authors write, "Targets in three regions…. Were SATB1+… but CCK-…" They should explain what this means, if anything. Does this point toward a particular type of cell? Another example of this is in subsection “MSDB neurons immunopositive for PV and/or SATB1 and/or mGluR1a”, where the authors write, "The largest group comprised PV-/SATB1+/mGluRa- neurons". What type of cells are these? Do they correspond to cells that have been reported previously?

We thank the reviewer for this important point about neuronal identities and we have revised our statements for clarity. To our knowledge, there is no previous report on mGluR1a and SATB1 expression in different combinations with PV in the medial septum. Previous classification has been limited to PV+/PV- or GAD+/GAD- (e.g. Simon et al., 2006), and also HCN1 immunoreactivity along the somatodendritic membrane of PV+ and PV- medial septal neurons (Varga et al. 2008). But this latter classification did not lead to a definition of cell types since the axons were not recovered. We have reported SATB1 in some medial septal neurons in the rat (Viney et al., 2003) and provide other novel marker combinations of recorded and labelled neurons in Table 4. We have now revised the first part of the Results section:

“Neuronal subpopulations in the MSDB can be defined by the expression of different molecules (Wei et al., 2012), and combinational expression profiles help define distinct cell types (Viney et al., 2013). We observed that metabotropic glutamate receptor 1a (mGluR1a), along with the transcription factor SATB1 (Huang et al., 2011), show differential immunoreactivity with parvalbumin (PV). As in the cortex, PV neurons in the MSDB represent a subpopulation of GABAergic neurons, but PV is expressed by many different kinds of neurons (Simon et al., 2006; Varga et al., 2008; Viney et al., 2013). GABAergic Teevra neurons are immunopositive (+) for PV and SATB1 but lack detectable immunoreactivity (–) for mGluR1a (Joshi et al., 2017).”

To our knowledge, no cell types in the presubiculum have been previously defined using SATB1 as one of the markers. Based on data from developmental studies of cortical SATB1 expression, we have updated the Results (lines 191-196): “Targets in three regions (PSd, RSg and FC) were SATB1+ (n=13/13; Figure 4B,E), but CCK– (n=0/10) (Table 6). The identity of SATB1+ neurons in these regions remain to be determined by visualising the processes of the cells, and are likely to include both PV+ and somatostatin+ interneurons (Nassar et al., 2015), given that SATB1 is highly expressed within these subpopulations in the cortex (Close et al., 2012; Denaxa et al., 2012)”

PV-/SATB1+/mGluRa- neurons are also undefined in the MSDB, and future work will elucidate their identities. We have updated this part of the Results to take this into account (subsection “MSDB neurons immunopositive for PV and/or SATB1 and/or mGluR1a”): “Apart from PV+/SATB1+/mGluR1‒ CA3-projecting Teevra cells (Joshi et al., 2017) and PV+/SATB1+/mGluR1a+ EC-projecting orchid cells, the identities of neurons within the other groups remain to be determined.”

4) With regard to Figure 2, the authors state that 5/8 neurons formed thin collaterals/terminals in deep layers of caudo-dorsal EC (subsection “Firing patterns of septo-entorhinal neurons”). What about the other 3 neurons?

We have now added (in subsection “Firing patterns of septo-entorhinal neurons”): “The main axons of the other 3 neurons faded before collateralization in the EC due to insufficient labeling (Table 1)”. However, the firing patterns, molecular profiles and in one case the other collaterals (AJ50j, Table 1), were similar to the 5 neurons with confirmed terminations in EC, which are reported in the subsequent Results sections.

5) The description of the dendrites of labeled MS neurons is anecdotal (subsection “GABAergic orchid cells project to multiple extra-hippocampal regions”). It would be strengthened by quantitative analysis and comparison to other populations of MS neurons.

We have counted the number and direction of primary dendrites from 4 orchid cells and 4 Teevra cells and added the following to the Results section:

“Orchid cells had 6.0 ± 1.8 primary dendrites (mean ± s.d., n=4 cells) extending from all axes, which were similar to the distributions of CA3-projecting Teevra cell dendrites (Joshi et al., 2017) (χ2 = 0.02, *P* = 0.8817, n=4 orchid cells versus n = 4 Teevra cells).”

6) The authors write, "We determined the molecular profiles of 13 targeted cortical neurons and used electron microscopy to test the reliability of predicting synaptic connections from light microscopy" (subsection “Preferential synaptic targets of orchid cells”). Please state from how many mice. This already seems a small number of representative neurons from which to draw a strong conclusion, and maybe even less so if from few mice.

The 13 identified synaptic junctions are on one neuron from 1 mouse (neuron TV58g, Table 6). We have updated this sentence accordingly. Analysis of the molecular profiles of targets was only possible for the most strongly labelled MS cell, since terminals were visible with fluorescence microscopy where targets could be tested with immunohistochemistry. Our previous data in rat and mouse CA3 and mouse retrosplenial cortex where we established that close apposition of medial septal GABAergic terminals using fluorescence microscopy predict with high confidence type II synaptic junctions with cortical interneurons (Joshi et al., 2017, Viney et al., 2013, Unal et al., 2015), supports our current predictions.

7) The authors write, "The axon showed target selectively, as nearby interneurons with different molecular profiles were not targeted" (subsection “Preferential synaptic targets of orchid cells”). Again, proper quantification is needed here.

We succeeded in identifying 13 targeted interneuron somata, all of which were SATB1 immunopositive in their nuclei and 10/10 tested were CCK-immunonegative (Table 6 and Figure 4). Only a subpopulation of MGE-derived interneurons show SATB1 expression, thus these data suggest that ‘orchid cells’ selectively innervate this subpopulation of interneurons. Full characterisation of GABAergic interneurons in the presubiculum and entorhinal cortex awaits further studies possibly using transcriptomic data (now stated in the Results section).

8) The introduction of 8 additional neurons in subsection “Behavioral and network state-dependent firing of orchid cells” raises additional questions that require clarification. What are the false positive and false negative rates for the preceding criteria? In other words, when these criteria are applied to all neurons with recovered axons (including those that have not been reported), how many meet the criteria and don't have EC projecting axons, and how many don't meet the criteria but do have EC projecting axons / other properties of 'orchid cells'?

We thank the reviewer for raising these important points. We have not yet recovered any neuron with an ‘orchid cell’ firing pattern that targets other areas of the temporal cortex. Likewise, neurons with different firing patterns target other areas e.g. the two septo-hippocampal/dentate cells included for comparison (subsection “Behavioral and network state-dependent firing of orchid cells”). The CA3-projecting Teevra cells (Joshi et al., 2017) did not target the dentate gyrus or other areas outside the hippocampus and had quantitatively different firing patterns to orchid cells. We have also added this sentence to the Results section: “We did not observe any non-EC projecting MS neurons (e.g. septo-hippocampal neurons) that exhibited all 4 features of the reported orchid cells (data not shown).”

Indeed, further labelling experiments will likely reveal additional cell types. For example, since the submission of the paper, we recently recorded a strongly-labelled theta trough-firing neuron with long bursts, also PV+, that targeted the RSg and CA1 but not the EC (unpublished data). It will take many months to years to obtain more of these neurons with a selective labelling targeting strategy based on differential firing patterns, as they are rarely encountered.

We state in the Materials and methods section under “in vivo extracellular recordings and juxtacellular labeling” that “animals with recorded neurons projecting to other parts of the temporal cortex and with firing patterns that differ from orchid cells will be reported in detail elsewhere”.

9) In subsection “Behavioral and network state-dependent firing of orchid cells”, it is unclear what the authors meant in the following passage: "The first peak was at 143.8[…] during locomotion[…] but occurred later during the other states". What exactly does "the first peak" refer to?

The peak refers to the peak in each autocorrelogram (highest counts); we have updated the text to state this.

10) In subsection “Behavioral and network state-dependent firing of orchid cells”, what is meant by "due to state-dependent features of the spike bursts"?

This has now been changed to: “We hypothesized that this increase in mean firing rate during locomotion was due to changes in the spike burst properties”

11) The results described in subsection “Behavioral and network state-dependent firing of orchid cells” are not entirely clear. At one point, the authors refer to "the identified CA3-projecting Teevra cells". Which ones? Do the authors mean the 2 cells that are mentioned in this subsection?

To assess the reliability of predicting axon terminations from firing patterns, we included 2 novel neurons that had long duration bursts like orchid cells but had a low burst incidence and projected selectively to the dentate gyrus and CA3. We also contrasted previously published Teevra cells, which project to CA3, but not the dentate gyrus (Joshi et al., 2017). We now state that the Teevra cells are ‘another group of neurons’ separate to the 2 additional (novel) neurons.

12) In the Materials and methods section, the authors write, "γ power was also used to estimate the location of the LFP recording". How exactly?

We now explain in the Materials and methods section as follows: “In addition to polarity of the SWRs, in three cases (AJ44j, AJ50h, AJ50j), mid γ (55-80 Hz; detection threshold: >1 s.d. above the mean cycle amplitude) coupling to the peak of theta oscillatory activity was also used to estimate the location of LFP recording in stratum pyramidale. Mid-γ (55 – 80 Hz) is coupled to the peak of CA1 pyramidale theta oscillatory activity (Colgin et al., 2009; Lasztóczi and Klausberger, 2014; Schomburg et al., 2014).”

13) In the Materials and methods section, the authors write, "If cross-frequency coupling was observed". How was this defined?

We used the Rayleigh test for uniformity of distribution between the troughs of γ and theta cycles to establish cross frequency coupling. Mean phase and mean vector length were used as measures of the preferred phase and coupling strength, respectively. We have now clarified the Materials and methods section as follows:

“Theta phase was calculated by linear interpolation between troughs of the band-pass filtered theta oscillations, with 0° and 360° set as the troughs. The Rayleigh test was used to test for uniformity of circular phase distributions. Mean phase and mean vector length were used as measures of the preferred phase and coupling strength, respectively, both for the spike-theta and γ troughs-theta coupling. Mid-γ (55-80 Hz) troughs were detected from the CA1d LFP for the entire recording period (detection threshold: >1 s.d. above the mean cycle amplitude), with most γ troughs detected during movement-related theta oscillations. If cross-frequency coupling was observed (γ troughs coupled to theta peak), spike coupling was measured for γ cycles as described for theta above. Phase histograms were smoothed by convolving with a Gaussian.”

14) In the Materials and methods section, the authors write, "Inter-burst intervals were calculated by taking the timing of the first spike of each burst". This method explanation does not make sense.

We now write: “Interburst intervals were calculated by measuring the time elapsed between the first spikes of consecutive bursts.”

15) The experiment described in Figure 6 (subsection “Relationship between entorhinal and hippocampal oscillations”) is quite confusing. It states that 1 pyramidal cell was recorded in EC to confirm the recording location – does this imply that the data reported is only from 1 mouse? When talking about MS firing rate distributions being non-Poisson, is this referring to 9/9 neurons from the same mouse? More generally, the point of these experiments is unclear. How do they go beyond similar experiments from the Buzsaki lab (e.g. Chrobak and Buzsaki, 1998; Mizuseki et al., 2009)?

Data from the Buzsaki lab are from rats, and it is difficult to directly compare published data of their recording locations within the EC to the site of medial septal orchid cell terminals within the mouse EC. We have now completed revised this part and provide an updated Figure 6 with additional data based on experiments with 7 more head-plate implanted mice. Our aim was to determine whether the rhythmic network activity that we measure in the hippocampal CA1 pyramidal cell layer (our ‘reference LFP’) is similar to that at the site of orchid cell terminals, since orchid cells do not project to CA1. For all recordings we consistently use the CA1 LFP to analyse spike timing, because in CA1 we have a consistent comparison reference point in the pyramidal cell layer, particularly in relation to theta cycles.

In the revised subsection “Relationship between dorsal hippocampal and caudal entorhinal rhythmic activity”, we report that the caudo-dorsal EC, in mouse, has similar LFP rhythmic activity to CA1, based on the strong coupling of EC theta troughs to the troughs of CA1d theta troughs. This was achieved by recording LFPs from at least 2 sites within the caudo-dorsal EC simultaneously with CA1d strata pyramidale/oriens LFPs (Figure 6). The locations were accurately determined by juxtacellularly labelling principal neurons or interneurons and measuring the relative depth. These difficult experiments were successful in 4 mice; 4 other mice had neurons labelled outside the EC and were excluded from analysis. We have updated the Results section and Materials and methods section accordingly.

We have revised the part on MS neuron firing distributions around sharp-wave ripples to state that it includes n=4 orchid cells from 4 mice and n=5 putative orchid cells from 4 mice.

16) The data in Figure 7 would be more useful if they could be more directly compared to the cell labeling data. For example, is the proportion of triple positive neurons similar to the proportion of 'orchid cells' in the total population of recorded MS neurons? And, can the data in Figure 1fFbe normalized to the total cell populations in Figure 7?

Our recording strategy was based on the location of the first three recovered ‘orchid’ cells in a limited to a part of the medial septum, whereas the stereological data characterises the entire MSDB in order to provide a baseline for all studies (now stated in Results section). At this stage, we do not yet know if the molecular profile of orchid cells, which is different from Teevra cells, may include other GABAergic neurons with different axonal target areas (stated in the Discussion section).

17) Overall the organization of the manuscript makes it harder to follow than it perhaps needs to be. For example, would it be clearer if when describing the labeled neurons if their anatomy and molecular markers were described first, and then the electrophysiology was built upon from this? At present, the manuscript jumps back and forth between anatomy and electrophysiology.

We consider it important to show results of anatomical and electrophysiological analysis together because the data are obtained from the same neurons and we would like to emphasize the biological unity of the cells. We departed from this in the first and last figures that provide supporting evidence of the EC-projecting subpopulation of MSDB neurons. This is consistent with our previous efforts in rats e.g. reviewed in Somogyi et al., (2014).

[Editors' note: further revisions were requested prior to acceptance, as described below.]

1) In the Introduction, "such as a place cell sequences" should be "such as place cell sequences".

This has now been revised.

2) In subsection “Medial septal neurons projecting to the entorhinal cortex”, add "for PV, SATB1, and mGluR1" after "triple immunopositive" to avoid confusion.

Revised.

3) In subsection “GABAergic orchid cells project to multiple extra-hippocampal regions”, the authors write, "mid-γ oscillations were coupled to theta cycles preferentially at the theta peak (n = 7/8 neurons)". This is confusing because γ was not recorded intracellularly in this study. So, clarification is needed to explain why measurements are given with regard to numbers of neurons.

Changed to: “As expected, mid-γ oscillations were coupled to theta cycles preferentially at the theta peak (detected from CA1d LFPs of n=7/8 animals with recorded septo-entorhinal neurons, Figure 2G).”

4) In subsection “Behavioral and network state-dependent firing of orchid cells”, "different in rates" should be "different than rates".

This sentence now reads: “In addition to behavioral state-dependent differences in rhythmic firing, the mean firing rates of orchid cells during locomotion were different than rates during immobility”

5) In subsection “MSDB neurons immunopositive for PV and/or SATB1 and/or mGluR1a”, the authors use the word "inhibited", but it would perhaps be better to use a different word choice because "inhibited" implies that IPSPs or IPSCs were actually measured.

Changed to: “with 4/9 neurons reducing their firing rate during SWRs”.

6) In the Discussion section, the authors write, "orchid cells represent a distinct subpopulation of Komal cells", but this seems incorrect because the authors stated that orchid cells fire at the theta peak and Komal cells fire at the theta trough.

We thank the Editors for this crucial point. Teevra and Komal cell groups were clustered independently of preferred theta phase. Most Komal cells fire at the peak of CA1d theta oscillations, but there is a subpopulation that fire at the theta trough (Figure 1D in Joshi et al., 2017). Komal cells were not labelled in the Joshi et al., 2017 study. We now write:

“Juxtacellular labeling of Teevra cells revealed that they targeted CA3 but no extra-hippocampal areas. These CA3-projecting neurons preferentially fired at the trough of CA1d theta oscillations. Komal cells were not labeled but most preferentially fired around the peak of CA1d theta oscillations. In contrast to Teevra cells, rhythmically bursting orchid cells, reported here, increase firing during locomotion, have long duration bursts, fire preferentially at the peak of CA1d theta oscillations, and innervate the PrSd and EC. Thus, orchid cells represent a distinct subpopulation of Komal cells defined by their theta phase firing preferences and synaptic target regions. The other subpopulation of Komal cells that preferentially fire at the theta trough (Joshi et al., 2017) remain to be defined using juxtacellular labeling in awake animals, as we have focused on theta peak firing cells for the very difficult labeling experiments in the present study.”

7. In the Discussion section, the authors should add, "in the present study" at the end of the sentence for the purpose of clarity.

We have now written: “as we have focused on theta peak firing cells for the very difficult labeling experiments in the present study.”

8) In the Materials and methods section, the authors should replace "elsewhere" with "in planned future studies".

Now revised as: “Animals with recorded neurons projecting to other parts of the temporal cortex and with firing patterns that differ from orchid cells will be reported in detail in planned future studies”

9) In the Materials and methods section”, "remove" should be "removal".

Revised.

10) In the Materials and methods section, providing more detail (i.e., the width of the wavelet) would be helpful.

We now provide further details: “Time-frequency plots (Figure 5—figure supplement 1) used the ContinuousWaveletTransform in Mathematica (Morlet wavelet, 1 kHz sampling rate, 12 octaves, 16 voices).”

11) In Table 2, it is unclear what the letters a-k mean. This should be explained in the caption.

We have now included in the legend: “Letters refer to individually recorded neurons. E.g. in animal TV58, recorded neurons ‘e’ and ‘g’ were selected for juxtacellular labeling; in animal TV50 only neuron ‘a’ was recorded and labeled.”